# MambaPEFT: Exploring Parameter-Efficient Fine-Tuning for Mamba

**Masakazu Yoshimura**[*]**, Teruaki Hayashi**[*] **& Yota Maeda**
Sony Group Corporation
Japan
{masakazu.yoshimura, teruaki.hayashi, yota.maeda}@sony.com

## Abstract

An ecosystem of Transformer-based models has been established by building large models with extensive data. Parameter-efficient fine-tuning (PEFT) is a crucial technology for deploying these models to downstream tasks with minimal cost while achieving effective performance. Recently, Mamba, a State Space Model (SSM)-based model, has attracted attention as a potential alternative to Transformers. While many large-scale Mamba-based models have been proposed, efficiently adapting pre-trained Mamba-based models to downstream tasks remains unexplored. In this paper, we conduct an exploratory analysis of PEFT methods for Mamba. We investigate the effectiveness of existing PEFT methods for Transformers when applied to Mamba. We also modify these methods to better align with the Mamba architecture. Additionally, we propose new Mamba-specific PEFT methods that leverage the distinctive structure of Mamba. Our experiments indicate that PEFT performs more effectively for Mamba than Transformers. Lastly, we demonstrate how to effectively combine multiple PEFT methods and provide a framework that outperforms previous works. The source code is available at: https://github.com/sony/mambapeft.

## 1 Introduction

Modern large-scale models, also known as Foundation Models, are heavily based on Transformers (Vaswani, 2017). Transformer-based pre-trained models span diverse domains such as language, vision, and multi-modal applications. Despite their widespread use, Transformers have a notable drawback: their computational inefficiency with long sequences. The computational complexity of the attention module scales with the square of the sequence length.

To address this fundamental drawback, Gu & Dao (2023) proposed Mamba, a linear-time sequence model that leverages the strengths of State Space Models (SSMs). While Transformers are constructed from attention modules, Mamba is based on the SSM architecture, allowing it to handle long sequences more efficiently. Additionally, Mamba has been shown to perform better than Transformers with the same number of parameters on major tasks such as natural language processing (NLP) (Gu & Dao, 2023) and computer vision (CV) (Zhu et al., 2024). This fact makes Mamba stand out from other post-Transformer architectures with sub-square time complexity (Peng et al., 2023; Sun et al., 2023). Mamba-based models have been proposed across a wide range of domains (Behrouz & Hashemi, 2024; Liang et al., 2024; Zhang et al., 2024b; Li et al., 2024). Mamba has the potential to go beyond the Transformer ecosystem.

Parameter-efficient fine-tuning (PEFT) is essential for adapting such large-scale models to downstream tasks. Fine-tuning all parameters of these models results in high computational costs. PEFT enables additional training for large-scale Transformers with limited computing resources and data. Early examples of PEFT include the fine-tuning of pre-trained language models for NLP tasks (Houlsby et al., 2019; Hu et al., 2021). Subsequently, it has been extensively adopted across a wide range of applications (Yeh et al., 2024; Wang et al., 2023; 2024a). While many PEFT methods have been extensively studied for Transformers (Lialin et al., 2023; Han et al., 2024), research on PEFT methods for Mamba remains limited.

---

[*]Equally contributed

In this paper, we provide an exploratory and comprehensive investigation of PEFT for Mamba. First, we adapt representative PEFT methods used in Transformers to the Mamba architecture and conduct extensive experiments. We also propose new PEFT methods specific to the Mamba architecture.

In the experiments, we benchmarked Mamba using PEFT methods, including seven main methods and a total of 20 derived variations (see Figure 1). Our benchmarks indicate that PEFT is more crucial for Mamba than for Transformers, with several methods outperforming the PEFT methods used for Transformers. Additionally, we demonstrate that these PEFT methods can be combined to surpass the performance of individual methods. We propose an efficient search technique to identify optimal PEFT combinations and hyperparameters. Unlike existing works that focus on specific high-performance methods, we explore a wide variety of PEFT methods. This exploration reveals suitable PEFT combinations and shows that merely combining high-performing methods is not sufficient.

The main contributions are as follows. First, to the best of our knowledge, we perform the first extensive and comprehensive benchmarking of PEFT methods for Mamba, including proposed Mamba-specific methods that are distinct from all PEFT methods for Transformers. Second, we propose a framework for achieving higher performance by combining multiple PEFT methods, which are obtained through our efficient search technique. Third, our results indicate that PEFT is more effective for Mamba than for Transformers, and several Mamba-specific phenomena are discovered through the experiments.

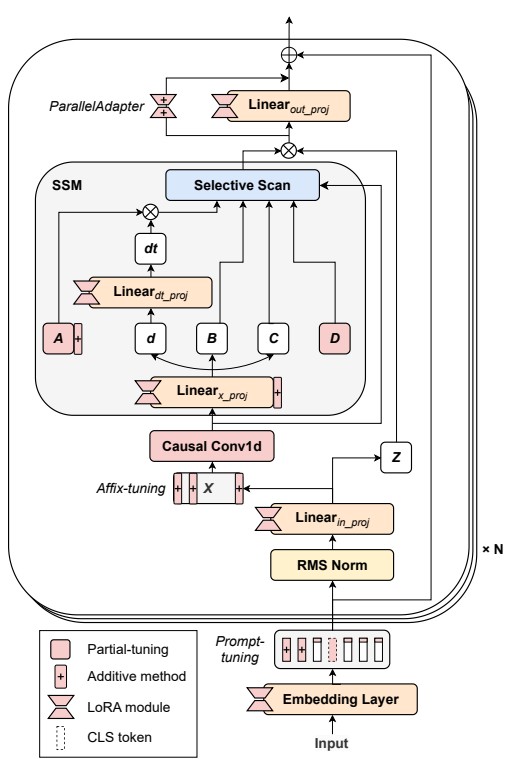

Figure 1: An overview of our proposed MambaPEFT. We investigate, improve, and propose 20 variations of seven PEFT methods for Mamba and search for the best combination.

## 2 RELATED WORK AND PRELIMINARY

In this section, we review related work and present the motivation behind our research. We begin with an introduction of Mamba. Subsequently, we discuss PEFT methods for Transformers. Finally, we highlight that research on PEFT for Mamba is limited.

### 2.1 MAMBA

SSMs are powerful methods for modeling time-dependent dynamic systems. It was developed and analyzed in detail to address the signal processing technique classically known as the Kalman filter (Kalman, 1960) and has been used to date in a wide range of fields, such as control engineering (Maciejowski & Huzmezan, 2007). An SSM (Gu et al., 2021a;b) takes an input $X_t$, converts it to a hidden state $H_t$, and then outputs $Y_t$. This procedure at each time step $t$ is

$$H_t = A_t H_{t-1} + B_t X_t, \quad Y_t = C_t H_t + D_t X_t. \tag{1}$$

As this model describes the relationship between continuous quantities, it must be described to deal with discrete quantities such as NLP. During step size $d$, the input is assumed to be invariant and the well-known theory of zero-order hold is applied. Putting

$$\overline{A} := \exp(dA), \quad \overline{B} := (dA)^{-1}(\exp(dA) - I)dB, \quad \overline{C} := C, \quad \overline{D} := D \tag{2}$$

allows us to obtain the final discrete representation:

$$H_t = \overline{A} H_{t-1} + \overline{B} X_t, \quad Y_t = \overline{C} H_t + \overline{D} X_t. \tag{3}$$

While many SSM-based DNN have tuned the A, B, C, and D (Gu et al., 2020; 2021b), *Selective State Space Model (S6)* was introduced in Mamba (Gu & Dao, 2023), in which the A, B, C, and D are dynamic and data-dependent as follows:

$$A_t = \exp(-\Delta_t A),\ B_t = \Delta_t W_B X_t,\ C_t = W_C X_t,\ D_t = W_D X_t \qquad (4)$$

where $\Delta_t = \mathrm{softplus}(W_\Delta(W_X X_t) + b_\Delta)$ for any $t$, $W_\Delta$. The $W_X$, $W_B$, $W_C$, $W_D$, and $A$ are trained matrices, and $b_\Delta$ is a trained bias.

Impressive results of Mamba in NLP have inspired researchers to adapt it to visual tasks. Vim (Zhu et al., 2024) is a pioneering work in this area. Similar to ViT (Dosovitskiy et al., 2020), it divides images into patches and then inputs the patch sequences into SSMs. Vim improved performance in visual tasks by processing patches bidirectionally with S6 and placing a class token in the middle of the patch tokens instead of at the beginning. There are other Mamba-based methods for visual tasks. For example, VMamba (Liu et al., 2024b) is a hybrid approach combining Mamba with 2D Convolutions. MambaVision (Hatamizadeh & Kautz, 2024) further integrates Attention. In this paper, we conduct experiments with Vim, which is a variant of the vanilla Mamba.

## 2.2 PARAMETER-EFFICIENT FINE-TUNING FOR TRANSFORMER

PEFT methods have been actively studied due to the large model size of Transformers extensively in both the NLP and CV communities. These studies can be categorized into four methods; partial, additive, reparametrization, and hybrid according to Lialin et al. (2023); Han et al. (2024).

**Partial-tuning methods.** While usual fine-tuning updates all parameters in the network, Partial-tuning updates a part of them. BitFit (Zaken et al., 2021) is a successful method in this category which tunes only the bias terms of the linear layers in Transformers. However, most of the linear layers in Mamba do not have the bias terms, which implies that it cannot be applied a priori. Other methods use pruning to decide where to tune Zhang et al. (2024c); He et al. (2023).

**Additive methods.** Additive methods add a small number of additional parameters or small networks to be fine-tuned. They can be further divided into additive adapter and additive token methods. The former adds adapter modules whose rank $r$ is reduced from the input dimension $d$ ($r << d$) by a bottleneck structure. Adapter (Houlsby et al., 2019) attaches the adapter in series with the feed-forward network (FFN) module in Transformers. In the following methods, Adapter+ (Steitz & Roth, 2024) improves the position of the input feature acquisition to the adapter. ParallelAdapter (He et al., 2022) and AdaptFormer (Chen et al., 2022) improve performance by inserting the adapter in parallel to the FFN. We adopt ParallelAdapter for Mamba because it is successful in both language (He et al., 2022) and vision (Chen et al., 2022) tasks. It attaches the adapter

$$x'_\ell = \mathrm{FFN}(x_\ell) + s \cdot \mathrm{ReLU}(x_\ell \cdot W_{\mathrm{down}}) \cdot W_{\mathrm{up}}, \qquad (5)$$

where the second term is the adapter with additional parameters $W_{\mathrm{down}} \in \mathbb{R}^{d \times r}$ and $W_{\mathrm{up}} \in \mathbb{R}^{r \times d}$, and a scaling hyperparameter $s$. In the latter category, Prompt-tuning (Liu et al., 2021) (resp. Prefix-tuning (Li & Liang, 2021)) adds learnable soft tokens to the input of the network (resp. soft tokens inside each Attention layer). Successive methods improve how to embed soft tokens (Liu et al., 2022; Razdaibiedina et al., 2023; Zhang et al., 2023). Though prompt-tuning for ViT adds learnable tokens at the beginning of the input (Jia et al., 2022; Wang et al., 2024b), it is unclear where the token should be added for Mamba because the order of tokens makes sense in SSM. In fact, Vim (Zhu et al., 2024) adds a class token in the middle of the input sequence. Hence, we investigate multiple choices of soft token insertion.

**Reparameterization methods.** Although reparameterization methods add low-rank adapters to Linear layers similar to the additive adapter methods, they adopt adapters that can be merged into the weights of the original Linear layer at the inference time. Due to this restriction, activation cannot be used in the adapter, while it eliminates the inference time overhead. The most notable one is LoRA (Hu et al., 2021), a method that reparameterizes the pre-trained Linear weights $W$ as $W' = W + s \cdot W_{\mathrm{down}} \cdot W_{\mathrm{up}}$. While subsequent studies have improved the efficiency by finding ways to reparameterize the weights (Lian et al., 2022; Hayou et al., 2024; Liu et al., 2024a; Jiang et al., 2024), we first investigate how many ranks of the adapter should be attached to which linear weights in Mamba. In this regard, we will use a simple LoRA with few hyperparameters.

**Hybrid methods.** Hybrid methods use multiple PEFT methods. While many approaches manually combine these methods (Mao et al., 2021; He et al., 2022; Karimi Mahabadi et al., 2021), several

methods automatically tune the combination of several PEFT methods. $S_4$ (Chen et al., 2023) utilizes random search while changing the dimension of the search space. AutoPEFT (Zhou et al., 2024) and NOAH (Zhang et al., 2024a) train a supernet containing all possible PEFT methods and discover the optimal combination using an evolutionary algorithm.

Supernet-based methods can only be used to search for each task individually. However, we aim to find the optimal combination for Mamba that can be generalized across multiple tasks, making it useful for various future applications. Hence, we propose an efficient search technique to find a compact combination of PEFT methods. This is necessary because we have many more PEFT methods and hyperparameters than previous works, resulting in a large search space.

## 2.3 PARAMETER-EFFICIENT FINE-TUNING FOR MAMBA

In contrast to the case of Transformers, there are limited studies on PEFT for Mamba. To the best of our knowledge, the only study in this context is Halloran et al. (2024), which applied the same rank of LoRA to all the linear layers in Mamba. With this motivation, we conduct an exploratory investigation and provide benchmarks of PEFT methods for Mamba.

## 3 INVESTIGATION OF PEFT METHODS FOR MAMBA AND BEYOND

In this section, we explore PEFT methods for the Mamba architecture. First, we discuss how existing PEFT methods for Transformers can be adapted to Mamba. Next, we present methods that have been improved and modified based on the characteristics of Mamba. Finally, we propose new PEFT methods specifically for Mamba and hybrid PEFT methods to search optimal combinations efficiently.

## 3.1 SIMPLE ADAPTATION OF EXISTING PEFT METHODS TO MAMBA

Some PEFT methods can be directly applied to Mamba because they are network architecture-independent. For example, **ParallelAdapter** attaches a parallel adapter to each FFN in Transformers. The counterpart of the FFN in Mamba is the out_proj layer, and hence we attach it to the *out_proj*.

**LoRA** is also a network architecture-independent method and can be applied directly to Mamba. However, it is still unclear which Linear layer should be targeted and what hyperparameter values should be used. We individually investigate LoRA on each module as **LoRA(embedding)**, **LoRA($in\_proj$)**, **LoRA($x\_proj$)**, **LoRA($dt\_proj$)**, and **LoRA($out\_proj$)** to clarify the appropriate way to apply LoRA to Mamba (see Figure 1).

## 3.2 RE-DESIGNING AND IMPROVING EXISTING PEFT METHODS FOR MAMBA

**Partial LoRA.** Mamba is a network characterized by numerous intermediate features with diverse properties, such as $X$, $Z$, $dt$, $B$, $C$, and so on (see Figure 1). In the previously mentioned normal LoRA, the inputs for multiple intermediate features are compressed together in the narrow rank of LoRA, even though they have different properties. Therefore, we now develop Partial LoRAs **LoRA$_p$($X$)**, **LoRA$_p$($Z$)**, **LoRA$_p$($dt$)**, **LoRA$_p$($B$)**, and **LoRA$_p$($C$)**, where we apply LoRA to only a part of the weights in the linear layer according to the output features. Since the dimensions of linear layers vary widely in Mamba (e.g. from 16 to 2048 in Mamba-1.3B), we also investigate whether there is an optimal dimension per layer or not to attach LoRA.

**Prompt-tuning.** Prompt-tuning can be applied directly to Mamba. As with Transformer, we simply add soft prompts to the input sequence. In normal Prompt-tuning, a prompt is inserted at the beginning of the input sequence. For Vim, however, we adjust the insertion position. This is because, unlike ViT, Mamba is a time-series model and works differently depending on where the prompt is inserted. In fact, Vim improved accuracy by inserting the class token in the middle of the sequence. In addition, Mamba Register (Darcet et al., 2024) found it is preferable to insert the register tokens at equal intervals. Hence, in this paper, we construct three prompt types; prefix (resp. infix, suffix) type, inserting prompt tokens at the beginning (resp. with equal intervals, at the end).

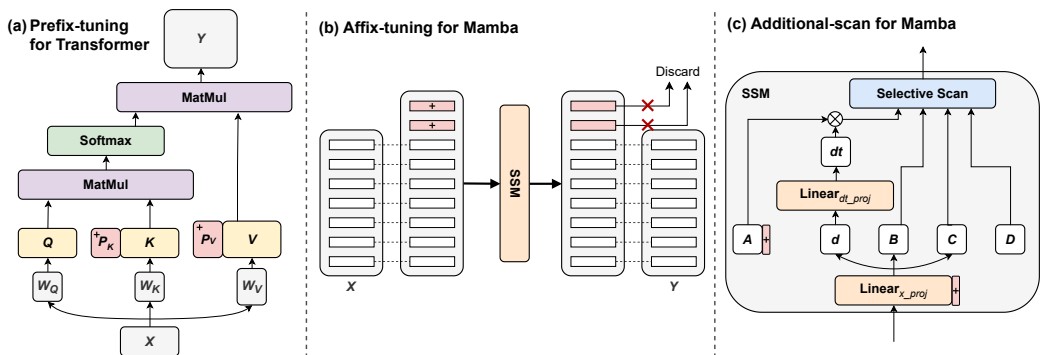

Figure 2: **(a)** Prefix-tuning designed for Transformer. It can't be applied to Mamba. **(b)** The proposed Affix-tuning, which we re-design for Mamba. It discards prefixes after SSM. This design allows us to insert affix tokens at arbitrary locations. **(c)** Additional-scan that we design for Mamba. In this method, we add a learnable dimension to the hidden state in SSM.

**Affix-tuning.** We cannot apply Prefix-tuning directly to time series models such as Mamba, since it is designed for the Attention mechanism based on query, key, and value. As shown in Figure 2, by adding soft tokens to only $K$ and $V$, Prefix-tuning successfully adds new information to all tokens in the input sequence without breaking the positional relationship of the original input sequence. Motivated by these observations, we propose a PEFT method that adds soft tokens before input to the SSMs and discards the output tokens of the inserted position (see Figure 2). This preserves the positional relationship between input and output tokens even when tokens are added. Furthermore, for Vim, we adjust the insertion position in the same way as for Prompt-tuning. Since the name Prefix-tuning can be misleading, we generalize by calling it *Affix-tuning*.

### 3.3 NEW PEFT METHODS DESIGNED FOR MAMBA

**Partial-tuning.** BitFit (Zaken et al., 2021) is effective among Partial-tuning methods because it fine-tunes bias terms with low degrees of freedom and they do not break the pre-trained model. Different situations for the Partial-tuning in Mamba is that, unlike Transformers, it does not utilize bias terms in linear layers without $dt\_proj$, while it has various other parameters with low degrees of freedom such as the parameters $A$, $D$, and the weights of the causal convolution whose dimension is one. Based on this situation, we construct *Partial-tuning* for Mamba, named **Bias-tuning**, **A-tuning**, **D-tuning**, and **Conv1d-tuning**, which fine-tune the previously mentioned parameters. For Vim, we also make **positional embedding** and **class tokens** learnable. In addition, we enhance our method from conventional Partial-tuning by modifying the weight decay. Specifically, instead of the usual weight decay of $|W|^2$, we propose applying a weight decay of $|W - W_{\text{pretrain}}|^2$ to preserve the pre-trained model.

**Additional-scan.** We propose a new efficient PEFT with fewer parameters by leveraging the SSM architecture. By increasing the state dimension of the SSM, we aim to store information in it to adapt to new tasks, called *Additional-scan*. It corresponds to additive methods. Specifically, we increase the state dimension of the input parameters of SSMs, $A$, $B$, and $C$, by $N'$. Eq. 3 then becomes as follows:

$$\overline{A}^{[T,L,N+\underline{\mathbf{N}'}]} = \exp(-\Delta^{[T,L,1]} \circ A^{[1,L,N+\underline{\mathbf{N}'}]}), \quad \overline{B}^{[T,L,N+\underline{\mathbf{N}'}]} = \Delta^{[T,L,1]} \circ B^{[T,1,N+\underline{\mathbf{N}'}]}, \quad (6)$$

where $T$, $L$, and $N$ are the token length, hidden size, and state dimension of the SSM, and $P^{[d_1,d_2,d_3]}$ represents $P \in \mathbb{R}^{d_1 \times d_2 \times d_3}$. Then, the core recurrent computation in SSMs becomes

$$\begin{aligned} H_t^{[L,N+\underline{\mathbf{N}'}]} &= \overline{A}_t^{[L,N+\underline{\mathbf{N}'}]} \circ H_{t-1}^{[L,N+\underline{\mathbf{N}'}]} + \overline{B}_t^{[L,N+\underline{\mathbf{N}'}]} \circ X_t^{[D,1]} \\ Y_t^{[L,1]} &= H_t^{[L,N+\underline{\mathbf{N}'}]} \cdot (C_t^\top)^{[N+\underline{\mathbf{N}'},1]} + D^{[L,1]}. \end{aligned} \quad (7)$$

In Mamba, a diagonal SSM is employed and each hidden dimension forms an independent recurrence. Hence, we can write down more about one hidden state dimension as follows;

$$
\begin{aligned}
[h_{t,1}, ..., h_{t,N}, h_{t,N+\underline{\mathbf{1}}}, ..., h_{t,N+\underline{\mathbf{N'}}}] = [\,&\exp(-\delta_{t,0}a_{0,1})h_{t-1,1} + \delta_{t,0}b_{t,1}x_{t,0}, ..., \\
&\exp(-\delta_{t,0}a_{0,N})h_{t-1,N} + \delta_{t,0}b_{t,N}x_{t,0}, \\
&\exp(-\delta_{t,0}\underline{\mathbf{a_{0,N+1}}})h_{t-1,N+\underline{\mathbf{1}}} + \delta_{t,0}\underline{\mathbf{b_{t,N+1}}}x_{t,0}, ..., \\
&\exp(-\delta_{t,0}\underline{\mathbf{a_{0,N+N'}}})h_{t-1,N+\underline{\mathbf{N'}}} + \delta_{t,0}\underline{\mathbf{b_{t,N+N'}}}x_{t,0}].
\end{aligned}
\tag{8}
$$

It can be observed that the added parameters do not affect the original hidden state, $h_{t,1}, ..., h_{t,N}$. This enables us to store new information with additional selective scan without affecting the pre-trained hidden memory and selective scan mechanism. To add new dimension to $A$, $B$, and $C$, we add additional trainable parameters to $A$ and to the Linear $\boldsymbol{x\_proj}$ that generates $B$ and $C$ (see Figure 2). The parameter $A$ in SSM needs careful initialization (Gu et al., 2020), known as the Hippo theory. Mamba uses S4D initialization (Gu et al., 2022), $A_{*,*,n} = n$, while in our case, the other $A$ values have already been changed by training, which implies that S4D initialization is not theoretically better. Thus, we propose an initialization of $A$ for Additional-scan. Specifically, the additional parameters of $A$ are initialized with the values of neighborhood pre-trained $A$. It is simple, while stable training is achieved. Accounting for eq. 8 and the pre-trained model use S4D initialization, the top of the state dimension is intended for long-term memory or selective scan from distant tokens, while the bottom is intended for short-term memory or selective scan from neighbor tokens. Thus, when fine-tuning, the suitable position of the additional state dimension is checked whether at the bottom as in eq. 8 or at the top.

### 3.4 Hybrid PEFT Search

We have introduced the primary seven methods as follows: ParallelAdapter, LoRA, Partial LoRA, Prompt-tuning, Affix-tuning, Partial-tuning, and Additional-scan. Specifically, there are five variations of LoRA, five variations of Partial LoRA, and six variations of Partial-tuning, making a total of 20 methods. This variety motivated us to propose a hybrid PEFT for Mamba, which combines multiple PEFT methods to enhance performance. There are three key elements to consider in our hybrid PEFT: the combination of PEFT methods, the hyperparameters of each PEFT method (e.g., a rank of LoRA), and the training hyperparameters for each method (e.g., learning rate). Exploring them simultaneously results in an enormous search space compared to the search space of previous works which target only several PEFT methods. Furthermore, the hyperparameters have a conditional search space, meaning they are only considered if the corresponding PEFT method is active. Thus, we propose a two-step approach that explicitly divides the problem into two parts to improve efficiency in the search process.

In the first step, we restrict the search space to combinations of PEFT methods with minimal trainable parameters within each method, i.e., we explore whether to activate each PEFT method with fixed hyperparameters with minimal trainable parameters. This phase is based on our observation that performance degrades when too many PEFT methods or trainable parameters are added. In the second step, based on the PEFT method combination suggested by the first step, we greedily search the hyperparameters of both PEFT methods and their training. Since this phase inevitably increases the number of trainable parameters, we also include an option to remove a specific PEFT method. This approach helps to prevent excessive parameter growth and allows us to allocate more parameters to the more critical methods. We utilize the TPE algorithm (Bergstra et al., 2011) implemented in Optuna (Akiba et al., 2019) for the search in each step. The detailed algorithm is provided in Appendix B.

## 4 Experiments and discussion

In this section, we present our experiments on 7 PEFT methods with 20 variations. We conduct experiments on both image and language tasks. Our methods are compared with state-of-the-art methods for Transformers. We provide detailed results and analysis based on these comprehensive experiments.

Table 1: Test accuracy on the VTAB-1k benchmark pre-trained on ImageNet-1K using the training technique of DeiT (Touvron et al., 2021). The best hyperparameter settings maximizing the averaged accuracy of six datasets written in the ablation studies are used for each PEFT method. The *Time Ratio* represents the ratio of the training time to the time taken for full fine-tuning. The number of trainable parameters is the average of all tasks.

| Model | Method | #Params (K) | Time Ratio | Natural | Specialized | Structured | Avg. |
|---|---|---|---|---|---|---|---|
| ViT-S | Scratch | 21,704 | | 10.66 | 56.12 | 24.83 | 26.20 |
| | Full | 21,704 | | 51.79 | 72.79 | 45.27 | 53.47 |
| | Linear Probing | 9 | | 60.87 | 78.13 | 30.57 | 51.74 |
| | FacT-TK | 16 | | 72.87 | 82.34 | 54.10 | 66.96 |
| | LoRA | 628 | | 73.60 | 82.22 | 57.61 | 68.68 |
| | Adaptformer | 333 | | 73.63 | 83.15 | 57.80 | 68.97 |
| | SPT-LoRA | 414 | | 74.75 | 84.75 | 56.99 | 69.38 |
| | Adapter+ | 122 | | 74.68 | 83.57 | 58.82 | 69.87 |
| Vim-S | Scratch | 25,450 | 1.00 | 8.33 | 49.87 | 28.16 | 25.42 |
| | Full | 25,450 | 1.00 | 59.35 | 68.74 | 34.39 | 47.08 |
| | Linear Probing | 9 | 0.19 | 62.50 | 77.25 | 31.97 | 52.75 |
| | CLS-token-tuning | 9 | 0.52 | 62.50 | 77.20 | 32.20 | 52.84 |
| | Pos-embed-tuning | 84 | 0.52 | 64.25 | 74.60 | 39.77 | 56.12 |
| | Bias-tuning | 37 | 0.54 | 68.94 | 79.16 | 45.05 | 61.03 |
| | D-tuning | 45 | 0.56 | 67.14 | 78.56 | 40.10 | 58.16 |
| | A-tuning | 598 | 0.54 | 72.01 | 80.72 | 49.59 | 64.40 |
| | Conv1d-tuning | 156 | 0.54 | 74.33 | 82.45 | 57.83 | 69.09 |
| | Prompt-tuning (w/o proj) | 12 | 0.53 | 65.40 | 78.51 | 38.35 | 56.77 |
| | Prompt-tuning | 307 | 0.54 | 69.92 | 79.20 | 47.75 | 62.54 |
| | Affix-tuning (w/o proj) | 230 | 0.59 | 72.26 | 81.03 | 50.73 | 65.04 |
| | Affix-tuning | 117,000 | 0.66 | 75.84 | 83.29 | 58.94 | 70.29 |
| | Additional-scan | 672 | 0.65 | 74.63 | 82.68 | 56.40 | 68.65 |
| | ParallelAdapter | 663 | 0.55 | 76.10 | 83.97 | 59.97 | 70.96 |
| | LoRA(embed) | 45 | 0.52 | 64.66 | 77.53 | 43.83 | 58.60 |
| | LoRA(x_proj) | 2,540 | 0.57 | 74.41 | 81.92 | 54.88 | 67.77 |
| | LoRA(dt_proj) | 2,442 | 0.57 | 75.35 | 83.05 | 57.12 | 69.30 |
| | LoRA(out_proj) | 2,663 | 0.57 | 76.42 | 83.96 | 60.08 | 71.12 |
| | LoRA(in_proj) | 1,483 | 0.67 | 76.58 | 84.08 | 60.16 | 71.25 |
| | LoRA$_p$(d) | 2,442 | 0.63 | 73.25 | 80.91 | 50.93 | 65.46 |
| | LoRA$_p$(C) | 2,417 | 0.57 | 72.78 | 81.57 | 51.35 | 65.61 |
| | LoRA$_p$(B) | 2,417 | 0.65 | 72.95 | 81.66 | 52.26 | 66.07 |
| | LoRA$_p$(Z) | 1,778 | 0.67 | 76.15 | 84.26 | 59.72 | 70.94 |
| | LoRA$_p$(X) | 1,778 | 0.66 | 76.64 | 83.89 | 60.84 | 71.52 |
| | All (w/ proj) | 119,765 | 0.86 | 74.67 | 82.96 | 53.92 | 67.68 |
| | All LoRA (r=8) | 1,228 | 0.80 | 76.11 | 84.32 | 59.92 | 71.02 |
| | LoRA(in_proj+out_proj) | 709 | 0.72 | 75.69 | 84.42 | 59.43 | 70.68 |
| | Hybrid (w/ proj) | 117,236 | 0.86 | **77.00** | 84.41 | **61.55** | **72.05** |
| | Hybrid (w/o proj) | 1,044 | 0.83 | 76.85 | **84.42** | 61.06 | 71.80 |

## 4.1 EXPERIMENTAL SETTINGS

**Models.** We use Vim-S (Zhu et al., 2024) as a base model in our experiments. We also experiment with ViT (Dosovitskiy et al., 2020) for comparison. We adopt pre-trained weights trained with ImageNet-1k (Deng et al., 2009) using DeIT (Touvron et al., 2021) training framework in all models.

**Dataset.** We conduct our evaluation on the VTAB-1k image classification dataset (Zhai et al., 2019). This dataset contains tasks from 19 domains. These tasks are grouped into three categories: Natural, Specialized, and Structured. For each task, 1000 images are used for training.

**Baselines.** We adopt LoRA (Hu et al., 2021), Adaptformer (Chen et al., 2022), FacT-TK (Jie & Deng, 2023), SPT-LoRA (He et al., 2023), and Adapter+ (Steitz & Roth, 2024) as existing PEFT methods for Transformers. For Vim, we evaluate the 20 PEFT methods explained in Section 3. As a baseline for Hybrid PEFT, we use a combination of all PEFT methods in the experiments.

**Implementation Details.** We follow the setup of Jie & Deng (2023) in our experiments, using AdamW optimizer (Loshchilov & Hutter, 2017) and training the model for 100 epochs. The learning rate is set to 1e-3, with a cosine scheduler and a warmup period of 10 epochs. A weight decay with 1e-4 magnitude is applied. We do not perform data augmentation. For ablation studies, we use the average accuracy on six tasks: CIFAR-100, Sun397, Camelyon, Retinopathy, Clevr-Count, and sNORB-Elev. In several previous studies, hyperparameters were tuned for each task, and the maxi-

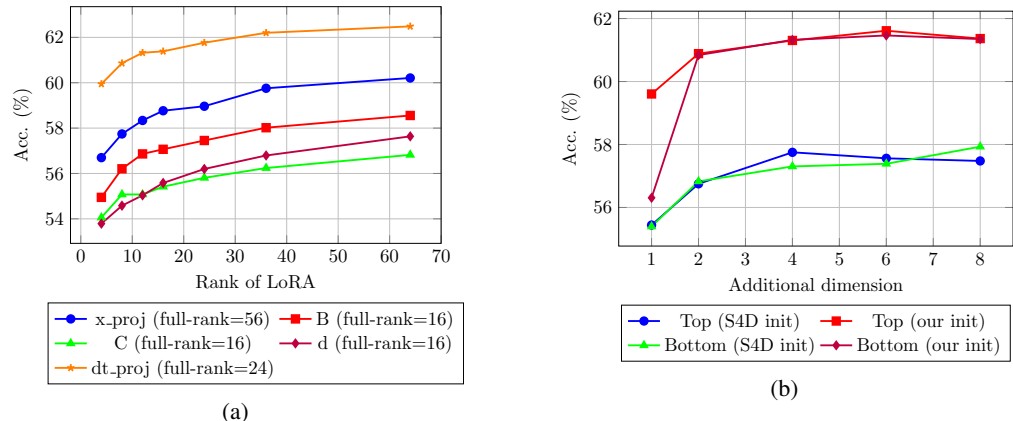

Figure 3: Ablation studies for **(a)** the relationship between LoRA rank and accuracy on small linear weights, and **(b)** the impact of additional dimensions in Additional-scan on performance.

Table 2: Ablation studies. **(a)** The position and number of tokens in Affix-tuning with projection. If the position is "Before", tokens are inserted before the in_proj layer. **(b)** The initialization method in Additional-scan. **(c)** The performance improvement in Partial-tuning with our proposed Weight Decay.

| Position | Type | #Tokens | Acc. (%) |
|----------|------|---------|----------|
| Before | Prefix | 1 | 23.85 |
| After | Prefix | 1 | 62.91 |
| After | Infix | 1 | 61.21 |
| After | Prefix | 2 | 63.36 |
| After | Prefix | 3 | **63.51** |
| After | Prefix | 4 | 56.57 |

(a)

| Location | Init. | Acc. (%) |
|----------|-------|----------|
| Bottom | S4D | 57.39 |
| | Const(0.1) | 61.25 |
| | Const(1) | 61.33 |
| | Const(5) | 60.05 |
| | Ours | 61.46 |
| Top | S4D | 57.56 |
| | Ours | **61.61** |

(b)

| Method | wd=1e-4 | wd=1e-3 | wd=1e-2 |
|--------|---------|---------|---------|
| Bias-tuning | +0.06 | +0.02 | **+0.17** |
| Conv1d-tuning | -0.40 | **+0.18** | -0.01 |
| Cls_token-tuning | -0.01 | **+0.08** | -0.01 |
| Pos-embed-tuning | **+0.18** | +0.09 | +0.02 |
| A-tuning | +0.07 | **+0.15** | +0.11 |
| D-tuning | -0.08 | **+0.02** | -0.04 |

(c)

mum validation accuracy during training was reported. In contrast, we tuned the hyperparameters to maximize the average of six tasks used in ablation, ensuring that the results obtained are universal and beneficial for many future studies. Additionally, the evaluation is conducted using the weights from the final epoch.

## 4.2 RESULTS FOR INDIVIDUAL PEFT METHODS

**Overall Results.** Table 1 shows the performance with hyperparameters that maximize accuracy. Many PEFT methods for Vim outperform the state-of-the-art ones for ViT. Interestingly, the PEFT methods for ViT start over-fitting early when the number of trainable parameters is increased and are unable to improve accuracy. On the other hand, Vim continues to improve its accuracy even with more trainable parameters. This is probably due to the highly modular structure of Mamba, which prevents its pre-trained memory from being corrupted by additional parameters. In addition, Vim has a greater performance improvement margin between full fine-tuning and PEFT than ViT has. These results suggest that PEFT is more effective for Mamba than Transformer.

**LoRA.** In our experiments, LoRA turns out to be one of the most effective PEFT methods with Vim. As shown in Table 1, PEFT methods perform well regardless of where LoRA is applied. The best performance is achieved with the partial LoRA, $LoRA_p(X)$. This result stems from our detailed investigation, indicating that the effectiveness of LoRA depends on the nature of the output features.

Another notable finding regarding LoRA is that when it is attached to a linear weight with a small dimension, the accuracy continues to increase even if the rank of LoRA exceeds the full-rank, as shown in Figure 3a. Full-rank refers to the smaller value of the input dimension or output dimension. This phenomenon is likely because the effective degrees of freedom are bounded by the full-rank. It follows that adding more parameters does not cause overfitting. Instead, it positively contributes to over-parameterization, similar to the findings in Ding et al. (2021). More details regarding the LoRA on large linear weights are described in Appendix Appendix A.1.

**Affix-tuning and Prompt-tuning.** The ablation studies on affix-tuning are shown in Table 2a. First, our Affix-tuning has multiple choices about where to add soft tokens because it is architecture-independent compared to Prefix-tuning for transformers, as shown in Figure 2. We test two positions, before or after the in_proj layer. If before, the output tokens are discarded just before the out_proj layer. If after, they are discarded after SSM. We find that it is important to affect additional tokens only on SSM by adding after the in_proj layer. Moreover, the best performance is achieved by adding an affix at the beginning of the input tokens, just as with Prefix-tuning for Transformer, and different from how to handle additional tokens for Vim in Zhu et al. (2024); Darcet et al. (2024). This should be because it is impossible to learn which positions are additional tokens rather than patch tokens unless they are trained from scratch. A detailed analysis of the positions and numbers of inserted tokens is provided in the Appendix A.1.

Note that these methods have the option of applying embedding projection to the prompts, which significantly increases the number of parameters. However, it can be merged during inference, resulting in zero computational cost increase. Our results show that embedding projection is effective in improving accuracy, and hence it is recommended to use it if memory allows.

**PEFT methods designed for Mamba.** We perform experiments on two newly proposed PEFT methods for Mamba. The results of Additional-scan are presented in Table 1. Additional-scan shows competitive performance with the best settings of LoRA and Affix-tuning, with fewer trainable parameters. Figure 3b and Table 2b list the experiments with different initialization for Additional-scan. It shows that our proposed initialization consistently outperforms the original S4D used in Mamba. The results of Partial-tuning are presented in Table 1. Performance highly depends on which components are made learnable. The highest performance is achieved by making conv1d learnable. Our proposed modified weight decay works effectively around 1e-3 strength (Table 2c). We analyze the number of trainable parameters and hyperparameters in Appendix A.2 in detail.

### 4.3 Results of Hybrid PEFT

The bottom of Table 1 shows the effectiveness of our two-step optimization approach. Combining all PEFT methods results in lower performance compared to using a single method. In contrast, our two-stage search method achieves higher performance with fewer parameters. These results indicate that selecting a preferred combination of PEFT methods is crucial.

We find that the high-performing single methods are not necessarily selected among the optimal combinations. An example of this combination is provided in Table 10 in Appendix B.2. We hypothesize that this phenomenon is related to the importance of ensuring model diversity in ensemble methods (Dietterich, 2000). Combining PEFT methods with different mechanisms can offset individual errors, mitigate overfitting, and enhance generalization performance. Detailed verification of this aspect remains a direction for future research.

### 4.4 Language tasks

In addition to the image tasks, we evaluate our method on language tasks using the vanilla Mamba Gu & Dao (2023). We experiment with a commonsense reasoning task, following the setup and dataset of Hu et al. (2023). This task consists of eight sub-tasks, which are evaluated on each task after training on a dataset that integrates training data from all tasks. We use Pythia (Biderman et al., 2023) as the Transformer baseline, which is pre-trained with the same dataset as Mamba. For a PEFT method for Mamba, we compare with SLL LoRA (Halloran et al., 2024). To the best of our knowledge, SLL LoRA is the only PEFT method for Mamba.

In the language tasks, the proposed Additional-scan is found to work as an efficient PEFT. We hypothesize that this is related to the amount of data. This experiment uses 170k datasets, in contrast to the 1k used for VTAB-1k. We hypothesize that learning a selective mechanism requires a relatively large amount of data, and Additional-scan works powerfully in such cases. In Affix-tuning, we find that, as the size of the base model increases, it achieves sufficient accuracy without the embedding projection. This is a valuable discovery because reducing memory costs is crucial for larger models. As discussed in Appendix A.4, we find that Additional-scan, Affix-tuning (w/o proj), and $LoRA_p(X)$ are all useful depending on the application and should be used accordingly.

Table 3: Experimental results of the commonsense reasoning tasks. Language Model Evaluation Harness (Gao et al., 2024) is used to benchmark. The best performance is highlighted in bold, and the second-best performance is underlined, excluding Full fine-tuning.

| Model | Method | #Params(%) | BoolQ | PIQA | SIQA | HellaSwag | WinoGrande | ARC-e | ARC-c | OBQA | Avg. |
|---|---|---|---|---|---|---|---|---|---|---|---|
| Pythia 160M | Full | 100 | 61.3 | 62.9 | 37.1 | 30.7 | 50.6 | 41.5 | 24.3 | 27.8 | 42.0 |
| | LoRA | 0.72 | 61.0 | 62.0 | 36.3 | 30.3 | 52.0 | 38.2 | 24.6 | 28.0 | 41.6 |
| Mamba 130M | Full | 100 | 56.1 | 65.3 | 38.7 | 35.3 | 52.0 | 46.4 | 25.7 | 32.8 | 43.8 |
| | SLL LoRA | 1.45 | 56.3 | 63.3 | 38.2 | 34.6 | 51.6 | 43.5 | 23.6 | 30.6 | 42.7 |
| | Additional-scan | 0.51 | 57.8 | 64.1 | 37.5 | 34.5 | 53.0 | 41.3 | 23.5 | 30.0 | 42.7 |
| | Affix-tuning (w/o proj) | 0.17 | 55.1 | 61.4 | 36.5 | 32.9 | 51.5 | 36.8 | 23.5 | 27.2 | 40.6 |
| | Affix-tuning | 64.64 | 59.7 | 64.3 | 38.2 | 35.2 | 51.9 | 42.9 | 24.0 | 29.0 | 43.2 |
| | LoRA(in_proj) | 2.23 | 53.5 | 62.9 | 38.2 | 33.8 | 53.1 | 46.4 | 23.7 | 30.8 | 42.8 |
| | LoRA$_p$(X) | 2.67 | 61.7 | 64.0 | 39.5 | 34.3 | 52.2 | 43.5 | 25.3 | 29.4 | 43.7 |
| Pythia 410M | Full | 100 | 55.0 | 68.4 | 42.1 | 40.8 | 53.9 | 50.8 | 26.7 | 30.0 | 46.0 |
| | LoRA | 0.77 | 61.3 | 67.7 | 40.8 | 39.2 | 54.9 | 48.1 | 24.7 | 28.6 | 45.7 |
| Mamba 370M | Full | 100 | 58.1 | 69.9 | 41.9 | 45.7 | 53.8 | 52.7 | 29.7 | 33.4 | 48.2 |
| | SLL LoRA | 2.30 | 59.5 | 69.6 | 42.2 | 44.1 | 54.9 | 50.6 | 26.3 | 30.8 | 47.3 |
| | Additional-scan | 0.47 | 61.9 | 69.3 | 41.2 | 45.3 | 54.9 | 28.4 | 49.5 | 31.4 | 47.7 |
| | Affix-tuning (w/o proj) | 0.16 | 62.0 | 67.7 | 39.3 | 46.3 | 54.1 | 47.8 | 28.2 | 31.0 | 47.0 |
| | Affix-tuning | 68.88 | 61.2 | 68.4 | 39.6 | 46.2 | 55.4 | 48.2 | 28.2 | 30.6 | 47.2 |
| | LoRA(in_proj) | 2.07 | 55.4 | 68.6 | 41.0 | 44.7 | 54.1 | 52.4 | 28.3 | 33.4 | 47.2 |
| | LoRA$_p$(X) | 2.67 | 60.8 | 68.8 | 42.1 | 44.7 | 56.2 | 50.4 | 27.4 | 32.2 | 47.8 |
| Pythia 1B | Full | 100 | 55.0 | 70.2 | 42.5 | 47.5 | 54.4 | 54.1 | 29.7 | 33.2 | 48.3 |
| | LoRA | 0.41 | 60.0 | 69.3 | 40.9 | 45.3 | 53.6 | 49.8 | 27.2 | 31.0 | 47.1 |
| Mamba 790M | Full | 100 | 62.0 | 72.1 | 44.8 | 54.0 | 55.9 | 57.7 | 31.2 | 35.2 | 51.6 |
| | SLL LoRA | 3.1 | 60.7 | 72.0 | 42.4 | 54.7 | 56.9 | 55.3 | 29.4 | 34.2 | 50.7 |
| | Additional-scan | 0.33 | 63.0 | 71.9 | 41.9 | 54.2 | 57.1 | 54.9 | 30.0 | 32.6 | 50.7 |
| | Affix-tuning (w/o proj) | 0.22 | 57.2 | 71.7 | 41.4 | 55.0 | 55.8 | 52.6 | 29.8 | 33.0 | 49.6 |
| | Affix-tuning | 69.99 | 61.0 | 72.5 | 41.0 | 54.9 | 55.6 | 54.6 | 29.6 | 33.8 | 50.4 |
| | LoRA(in_proj) | 1.47 | 61.7 | 71.9 | 44.0 | 50.8 | 56.7 | 56.3 | 30.5 | 33.8 | 50.7 |
| | LoRA$_p$(X) | 1.75 | 59.9 | 72.2 | 44.2 | 52.8 | 58.0 | 53.7 | 30.8 | 34.8 | 50.8 |
| Pythia 1.4B | Full | 100 | 58.6 | 71.1 | 42.7 | 53.6 | 55.1 | 58.5 | 29.9 | 34.8 | 50.5 |
| | LoRA | 0.44 | 60.1 | 71.3 | 42.5 | 50.1 | 58.9 | 57.6 | 29.6 | 33.6 | 50.5 |
| Mamba 1.4B | Full | 100 | 61.4 | 73.3 | 43.9 | 56.9 | 59.0 | 59.7 | 34.0 | 35.4 | 53.0 |
| | SLL LoRA | 4.64 | 59.7 | 73.5 | 43.1 | 56.9 | 60.7 | 59.7 | 31.7 | 36.0 | 52.7 |
| | Additional-scan | 0.26 | 63.0 | 73.5 | 42.8 | 57.5 | 60.5 | 60.9 | 32.4 | 37.4 | 53.5 |
| | Affix-tuning (w/o proj) | 0.09 | 63.6 | 74.0 | 41.9 | 58.9 | 60.6 | 61.6 | 33.4 | 36.8 | 53.9 |
| | LoRA(in_proj) | 1.13 | 62.6 | 73.6 | 43.7 | 55.6 | 59.7 | 58.3 | 31.7 | 35.6 | 52.6 |
| | LoRA$_p$(X) | 1.36 | 63.1 | 73.5 | 42.7 | 57.7 | 61.6 | 60.4 | 32.9 | 37.4 | 53.7 |

## 4.5 Limitations

This paper focuses on an exploratory investigation. Evaluating the applicability of our findings to larger models, such as vision models trained with ImageNet-21K (Ridnik et al., 2021) or large language models with billions of parameters (Touvron et al., 2023), is an area for future work. We release the source code to stimulate future research from communities.

## 5 Conclusion

We conducted a comprehensive investigation of PEFT for Mamba. Our investigation started by applying existing PEFT methods for Transformers to Mamba. We proposed Affix-tuning and Partial LoRA as modifications of existing PEFT methods and introduced Partial-tuning and Additional-scan as new Mamba-specific PEFT methods. We also provided an effective framework for combining them. These comprehensive experiments on image and language tasks led to several findings.

Our findings are as follows. (1) Mamba benefits from PEFT more than Transformers. It improves accuracy without overfitting, even with a larger number of parameters in PEFT (Compare Figure 4c and Figure 4d). (2) LoRA-based methods, particularly LoRA$_p$(X), are effective with limited data (See Table 1). (3) In contrast, the proposed Additional-scan excels with larger datasets (See Table 3 and Figure 7). (4) We found that initialization significantly impacts performance when adding an additional dimension to A, despite Mamba's robustness to initialization compared to other SSMs (See Figure 3b). (4) The proposed Affix-tuning was effective, especially for large Mamba models (See Table 3). (5) When adding LoRA on small linear weights, interestingly, over-parameterizing the original weight rather than decomposing it to a lower rank can be beneficial, which should hold for other architectures than Mamba (Figure 3a). (6) In hybrid PEFT, simply combining individually high-performance methods, as many previous works did, is inefficient. It is crucial to find the appropriate combination of PEFT methods. (See Table 1 and Table 9) We believe that these results make a valuable contribution to the Mamba research community.

## ACKNOWLEDGEMENTS

We would like to thank Takeshi Ohashi, Wei-Yao Wang, Helen Suzuki, and Julian Tanke for their helpful comments to this manuscript.

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

## A  MORE DETAILED ABLATION STUDIES

### A.1  INDIVIDUAL PEFT METHODS IN DETAIL

In the experiments, we need to determine the appropriate parameters for each method. To this end, we investigate the optimal settings by varying the degrees of freedom for each PEFT method. Specifically, the rank of LoRA ($r$), the number of additional tokens for Affix-tuning and Prefix-tuning ($n$), the number of dimensions for Additional-scan, and the strength of the proposed weight decay for Partial-tuning. Same as the ablation study in Table 2 and Figure 3 in the main text, we evaluate the averaged accuracy across six tasks: CIFAR-100, Sun397, Camelyon, Retinopathy, Clevr-Count, and sNORB-Elev.

First, each LoRA applied to large weights is evaluated. As shown in Figure 4d, the maximum accuracy with ViT can be obtained with approximately $r = 32$, whereas that with Vim can be obtained with a higher rank, as shown in Figure 4c. One reason is that the pre-trained Vim is less prone to collapse even when a large LoRA is added. Interestingly, the in_proj layer itself is prone to collapse and begins to degrade in accuracy at $r > 32$, while $LoRA_p(X)$ and $LoRA_p(Z)$, which apply LoRA to partial weights of the in_proj layer, are able to continue improving accuracy up to $r = 64$. By dividing LoRA for each output feature, the collapse is suppressed, and further accuracy improvement is possible with large parameters in LoRA. We have already discussed LoRA on small linear weights in the main text (Figure 3a)

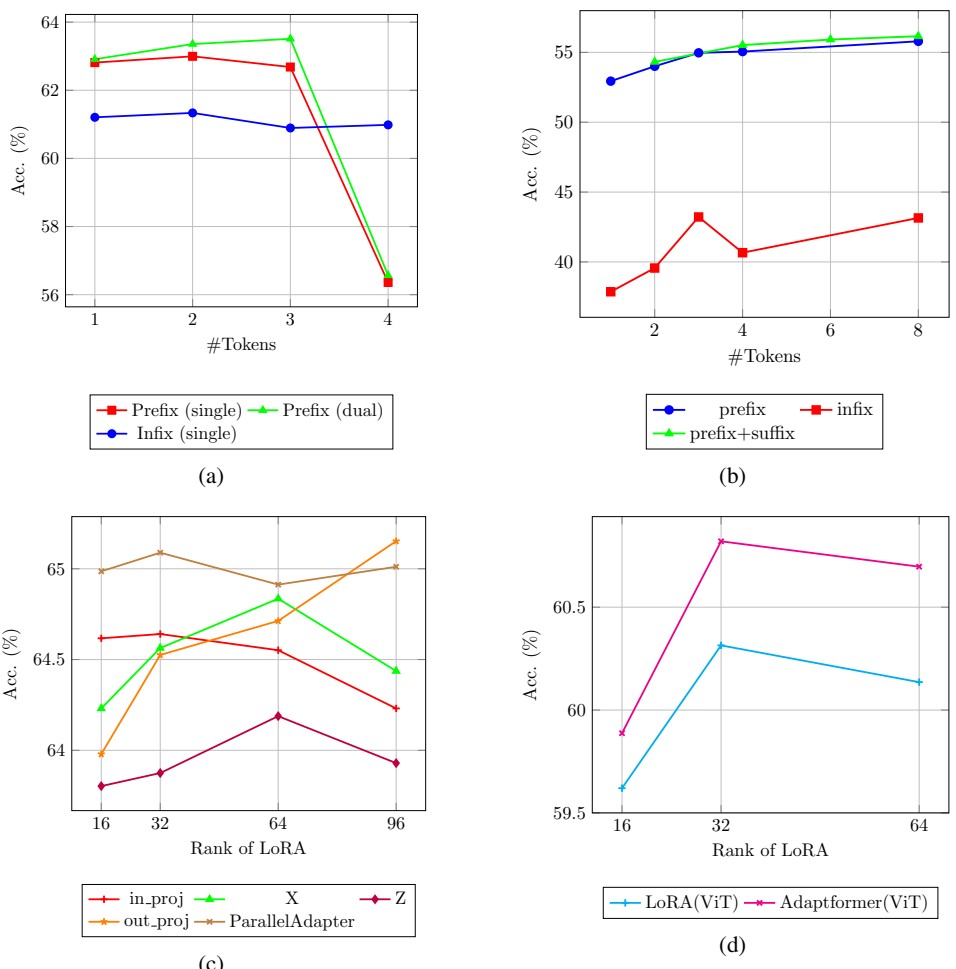

Figure 4: Ablation studies for vision tasks. **(a)** shows the relationship between the number of tokens and performance in Affix-tuning with projection. **(b)** shows the relationship between the number of tokens and performance in Prompt-tuning. **(c)** shows the relationship between the rank of LoRA and performance in Mamba. **(d)** shows the relationship between the rank of LoRA and performance in ViT. Performance consistently decreases as the rank increases up to 64, contrary to the case in Mamba.

For Prompt-tuning, adding soft tokens to both the beginning and end is the most effective, as shown in Figure 4b. Figure 4a also shows that the number of affixes improves performance up to 3 tokens. However, 4 or more affix tokens degrade performance due to over-fitting caused by increasing the number of parameters.

## A.2 EFFICIENCY OF PEFT METHODS FOR VISION MAMBA

In Table 1, we present a benchmark with the hyperparameter settings that maximize accuracy. As discussed in Appendix A.1, Vim is less prone to collapse and benefits more from increasing the number of training parameters in PEFT modules. Consequently, the number of trainable parameters for Vim in Table 1 is larger than that for ViT. Hence, we analyze the trade-off between accuracy and the size of PEFT modules. Specifically, all methods evaluated in Figure 3 and Figure 4 are plotted in Figure 5. The results indicate that the superior PEFT methods for Mamba achieve high accuracy even with a lower number of parameters and exhibit a better trade-off compared to those for ViT.

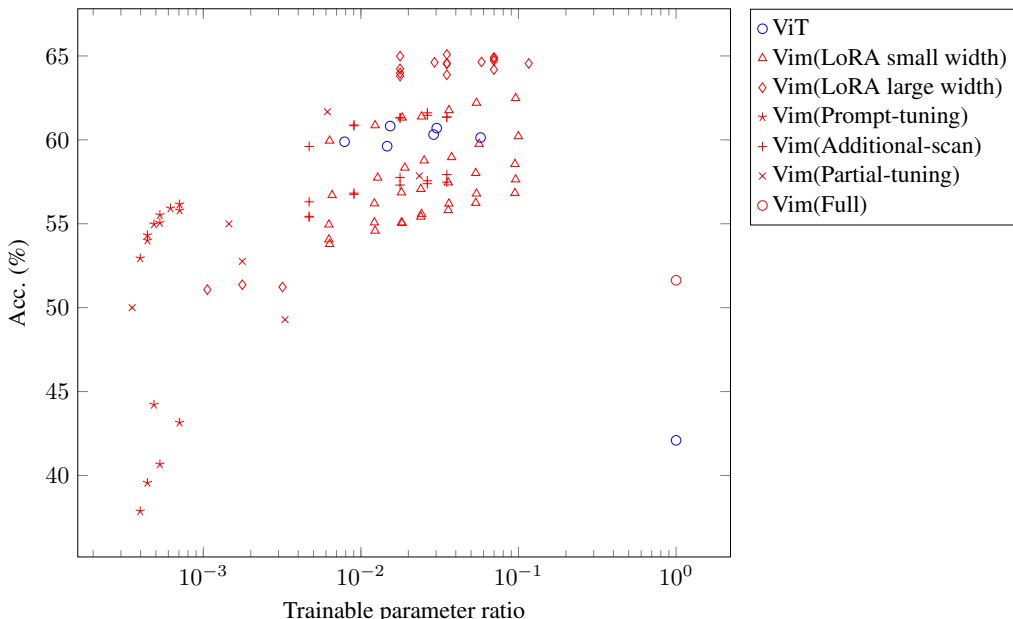

Figure 5: Scatter plot of the evaluations for the methods in Appendix A.1, showing the relationship between the proportion of trainable parameters and performance

## A.3    PER-TASK EVALUATION ON VTAB-1K

We present the results for each task in VTAB-1k, including different base model sizes and training data. Specifically, we also report the accuracy on other base model architectures such as Vim-B, Vim-tiny (Zhu et al., 2024), ViT-tiny (Touvron et al., 2021), and ViT-B (Dosovitskiy et al., 2020), as well as different pre-training datasets such as ImageNet-21k (Deng et al., 2009) or using the training framework of (Dosovitskiy et al., 2020). The information of Vim-B is recently released, so we additionally experimented the performance here in the Appendix. The compared PEFT methods for Transformers are LoRA (Hu et al., 2021), Adaptformer (Chen et al., 2022), Adapter+ (Steitz & Roth, 2024), FacT-TK (Jie & Deng, 2023), Head2Toe (Evci et al., 2022), VQT (Tu et al., 2023), BitFit (Zaken et al., 2021), SPT-LoRA (He et al., 2023), and LoSA (Mercea et al., 2024). The results are shown in Table 4.

Many PEFT methods for Vision Transformers have benchmarked with ViT-B pre-trained on ImageNet-21K. First, we clarify the accuracy change due to differences in our experimental settings and their settings. By using ImageNet-1K instead of ImageNet-21K, accuracy decreases by around 5 points. Changing the training framework to DeiT does not significantly affect accuracy. Reducing the model size from ViT-B to ViT-S results in an accuracy drop of about 2 points.

Next, we compare PEFT for Transformers and PEFT for Mamba using base models of a similar size. For small-sized base models, as mentioned in the main paper, we achieve a performance improvement of an additional 1.5 points over the state-of-the-art PEFT for Transformers. With even smaller models, such as ViT-tiny and Vim-tiny, it shows similar accuracy improvements. This confirms that our proposed method is effective for model sizes other than Vim-S used in the main paper.

Finally, we evaluate each task individually. PEFT for Vim outperforms PEFT for ViT on most tasks when compared within a similar base model size. Notably, PEFT for Vim is inferior to PEFT for ViT only in the dSpr-Loc task, which classifies the $x$-coordinates of objects of various colors, shapes, and sizes. We consider that one reason for this is that the attention mechanism can locate the objects more straightforwardly than Mamba. On the other hand, in the Clevr/count task, the accuracy with Vim is much higher than that with ViT. This task involves counting the number of objects in the image, and the recurrent nature of Mamba may be more suitable for counting the number of objects in the hidden state.

Table 4: Detailed test results on the VTAB-1k benchmark. "Mean" denotes the average accuracy for each category, and "Overall Mean" shows the average accuracy over 19 tasks. The (A), (T), and (D) stand for the network architecture, training framework, and dataset used for pre-training.

| Method | CIFAR-100 | Caltech101 | DTD | Flowers102 | Pets | SVHN | Sun397 | Mean | Camelyon | EuroSAT | Resisc45 | Retinopathy | Mean | Clevr-Count | Clevr-Dist | DMLab | KITTI-Dist | dSpr-Loc | dSpr-Ori | sNORB-Azim | sNORB-Elev | Mean | Overall Mean |
|---|---|---|---|---|---|---|---|---|---|---|---|---|---|---|---|---|---|---|---|---|---|---|---|
| **(A) ViT-B (T) ViT (D) IN21K** | | | | | | | | | | | | | | | | | | | | | | | |
| Full | 68.9 | 87.7 | 64.3 | 97.2 | 86.9 | 87.4 | 38.8 | 75.9 | 79.7 | 95.7 | 84.2 | 73.9 | 83.4 | 56.3 | 58.6 | 41.7 | 65.5 | 57.5 | 46.7 | 25.7 | 29.1 | 47.6 | 65.6 |
| Linear | 64.4 | 85.0 | 63.2 | 97.0 | 86.3 | 36.6 | 51.0 | 69.1 | 78.5 | 87.5 | 68.5 | 74.0 | 77.1 | 34.3 | 30.6 | 33.2 | 55.4 | 12.5 | 20.0 | 9.6 | 19.2 | 26.9 | 52.3 |
| BitFit | 72.8 | 87.0 | 59.2 | 97.5 | 85.3 | 59.9 | 51.4 | 73.3 | 78.7 | 91.6 | 72.9 | 69.8 | 78.3 | 61.5 | 55.6 | 32.4 | 55.9 | 66.6 | 40.0 | 15.7 | 25.1 | 44.1 | 62.1 |
| LoRA | 67.1 | 91.4 | 69.4 | 98.8 | 90.4 | 85.3 | 54.0 | 79.5 | 84.9 | 95.3 | 84.4 | 73.6 | 84.6 | 82.9 | 69.2 | 49.8 | 78.5 | 75.7 | 47.1 | 31.0 | 44.0 | 59.8 | 72.3 |
| FacT-TK | 70.6 | 90.6 | 70.8 | 99.1 | 90.7 | 88.6 | 54.1 | 80.6 | 84.8 | 96.2 | 84.5 | 75.7 | 85.3 | 82.6 | 68.2 | 49.8 | 80.7 | 80.8 | 47.4 | 33.2 | 43.0 | 60.7 | 73.2 |
| Adapter+ | 83.7 | 94.2 | 71.5 | 99.3 | 90.6 | 88.2 | 55.8 | 83.3 | 87.5 | 97.0 | 87.4 | 72.9 | 86.2 | 82.9 | 60.9 | 53.7 | 80.8 | 88.4 | 55.2 | 37.3 | 46.9 | 63.3 | 75.5 |
| LoSA | 82.5 | 92.8 | 76.1 | 99.7 | 90.5 | 82.0 | 55.8 | 82.8 | 86.6 | 97.1 | 87.0 | 76.7 | 86.9 | 81.5 | 62.3 | 48.6 | 82.1 | 94.2 | 61.7 | 47.9 | 45.6 | 65.5 | 76.4 |
| **(A) ViT-B (T) ViT (D) IN1K** | | | | | | | | | | | | | | | | | | | | | | | |
| Scratch | 7.6 | 19.1 | 13.1 | 29.6 | 6.7 | 19.4 | 2.3 | 14.0 | 71.0 | 71.0 | 29.3 | 72.0 | 60.8 | 31.6 | 52.5 | 27.2 | 39.1 | 66.1 | 29.7 | 11.7 | 24.1 | 35.3 | 32.8 |
| Full | 44.3 | 84.5 | 54.1 | 84.7 | 74.7 | 87.2 | 26.9 | 65.2 | 85.3 | 95.0 | 76.0 | 70.4 | 81.7 | 71.5 | 60.5 | 46.9 | 72.9 | 74.5 | 38.7 | 28.5 | 23.8 | 52.2 | 63.2 |
| Linear | 50.6 | 85.6 | 61.4 | 79.5 | 86.5 | 40.8 | 38.0 | 63.2 | 79.7 | 91.5 | 71.7 | 65.5 | 77.1 | 41.4 | 34.4 | 34.1 | 55.4 | 18.1 | 26.4 | 16.5 | 24.8 | 31.4 | 52.7 |
| Head2Toe | 54.4 | 86.8 | 64.1 | 83.4 | 82.6 | 78.9 | 32.1 | 68.9 | 81.3 | 95.4 | 81.2 | 73.7 | 82.9 | 49.0 | 57.7 | 41.5 | 64.4 | 52.3 | 32.8 | 32.7 | 39.7 | 46.3 | 62.3 |
| VQT | 58.4 | 89.4 | 66.7 | 90.4 | 89.1 | 81.1 | 33.7 | 72.7 | 82.2 | 96.2 | 84.7 | 74.9 | 84.5 | 50.8 | 57.6 | 43.5 | 77.2 | 65.9 | 43.1 | 24.8 | 31.6 | 49.3 | 65.3 |
| FacT-TK | 56.0 | 89.0 | 66.8 | 90.0 | 91.3 | 84.8 | 40.2 | 74.0 | 83.9 | 94.7 | 82.8 | 72.3 | 83.4 | 73.8 | 59.6 | 45.1 | 75.5 | 68.0 | 49.4 | 23.8 | 34.1 | 53.7 | 67.4 |
| LoRA | 52.7 | 89.1 | 66.4 | 90.8 | 90.7 | 85.4 | 40.0 | 73.6 | 84.6 | 94.7 | 82.6 | 73.2 | 83.8 | 76.6 | 62.5 | 50.0 | 80.0 | 75.8 | 48.2 | 27.8 | 36.7 | 57.2 | 68.8 |
| **(A) ViT-B (T) DeiT (D) IN1K** | | | | | | | | | | | | | | | | | | | | | | | |
| Linear | 35.4 | 86.5 | 61.8 | 83.9 | 88.3 | 43.4 | 37.7 | 62.4 | 80.0 | 90.8 | 74.3 | 73.4 | 79.6 | 40.6 | 37.3 | 36.7 | 59.6 | 20.3 | 25.0 | 16.6 | 24.2 | 32.5 | 53.5 |
| FacT-TK | 61.0 | 89.1 | 67.0 | 91.8 | 91.6 | 87.6 | 42.3 | 75.8 | 84.0 | 94.8 | 84.4 | 72.4 | 83.9 | 76.1 | 58.1 | 47.2 | 79.7 | 76.6 | 55.9 | 28.6 | 33.0 | 56.9 | 69.5 |
| LoRA | 61.3 | 89.5 | 68.2 | 92.9 | 91.3 | 88.4 | 42.8 | 76.3 | 84.0 | 95.0 | 85.0 | 73.2 | 84.3 | 79.4 | 58.8 | 49.5 | 81.7 | 81.0 | 54.2 | 31.2 | 38.1 | 59.2 | 70.8 |
| SPT-LoRA | 62.1 | 89.8 | 69.7 | 92.6 | 89.0 | 91.6 | 43.4 | 76.9 | 85.6 | 95.9 | 85.6 | 73.8 | 85.2 | 76.3 | 55.8 | 49.2 | 82.1 | 78.3 | 55.3 | 31.8 | 37.1 | 58.3 | 70.8 |
| Adapter+ | 61.6 | 91.0 | 68.6 | 92.5 | 91.8 | 89.8 | 43.2 | 76.9 | 85.5 | 95.1 | 85.3 | 73.7 | 84.9 | 78.4 | 58.8 | 50.7 | 81.2 | 83.6 | 55.1 | 32.7 | 35.6 | 59.5 | 71.3 |
| **(A) Vim-B (T) DeiT (D) IN1K** | | | | | | | | | | | | | | | | | | | | | | | |
| Full | 28.5 | 65.0 | 5.5 | 63.2 | 42.4 | 86.4 | 16.5 | 43.9 | 65.1 | 83.5 | 55.8 | 73.6 | 69.5 | 22.0 | 44.1 | 20.7 | 33.3 | 36.1 | 44.2 | 7.1 | 14.9 | 27.8 | 42.5 |
| Linear | 46.4 | 86.8 | 63.9 | 80.0 | 90.2 | 43.1 | 41.3 | 64.5 | 82.2 | 89.5 | 71.4 | 73.8 | 79.2 | 37.7 | 38.0 | 36.3 | 63.6 | 16.8 | 26.7 | 16.2 | 22.6 | 32.2 | 54.0 |
| Affix-tuning (w/o proj) | 59.0 | 88.1 | 66.8 | 84.5 | 90.8 | 52.5 | 43.0 | 69.2 | 79.3 | 93.5 | 82.2 | 73.8 | 82.2 | 62.0 | 54.3 | 43.5 | 76.8 | 59.9 | 50.7 | 25.2 | 33.8 | 50.8 | 64.2 |
| Affix-tuning | 62.2 | 88.8 | 68.3 | 91.2 | 91.3 | 88.9 | 43.7 | 76.4 | 83.0 | 94.1 | 83.5 | 73.5 | 83.5 | 76.4 | 59.1 | 33.7 | 79.2 | 79.6 | 50.8 | 31.1 | 35.3 | 55.6 | 69.1 |
| Additional-scan | 60.3 | 88.3 | 68.9 | 90.2 | 91.7 | 85.7 | 43.2 | 75.5 | 83.0 | 93.7 | 82.1 | 70.7 | 82.4 | 74.1 | 61.7 | 44.3 | 80.2 | 73.5 | 46.8 | 29.4 | 36.8 | 55.8 | 68.7 |
| ParallelAdapter | 62.8 | 89.8 | 68.4 | 91.5 | 91.2 | 88.2 | 44.0 | 76.6 | 83.5 | 95.0 | 85.0 | 72.4 | 84.0 | 83.3 | 64.2 | 50.2 | 79.3 | 78.9 | 53.0 | 33.8 | 40.6 | 60.4 | 71.3 |
| LoRA(out_proj) | 63.8 | 89.5 | 69.1 | 91.7 | 92.1 | 87.7 | 44.2 | 76.7 | 83.9 | 94.8 | 85.0 | 72.4 | 84.0 | 83.6 | 63.4 | 50.3 | 81.3 | 76.3 | 52.5 | 33.0 | 40.6 | 60.1 | 71.3 |
| LoRA(in_proj) | 63.7 | 90.2 | 67.7 | 91.0 | 90.8 | 89.7 | 43.9 | 76.7 | 85.0 | 95.1 | 84.6 | 72.5 | 84.3 | 81.3 | 60.8 | 50.8 | 79.9 | 82.2 | 50.9 | 31.4 | 37.6 | 59.4 | 71.0 |
| LoRA$_p$(X) | 63.9 | 90.1 | 68.4 | 91.2 | 91.0 | 90.4 | 43.9 | 77.0 | 84.1 | 95.2 | 84.0 | 72.5 | 83.9 | 83.0 | 62.7 | 49.6 | 79.9 | 81.7 | 51.4 | 31.2 | 32.9 | 59.9 | 71.3 |
| **(A) ViT-S (T) DeiT (D) IN1K** | | | | | | | | | | | | | | | | | | | | | | | |
| Scratch | 4.9 | 11.1 | 9.7 | 24.4 | 3.4 | 19.4 | 1.6 | 10.7 | 65.0 | 57.3 | 28.6 | 73.6 | 56.1 | 20.9 | 48.1 | 26.2 | 45.0 | 7.3 | 27.1 | 6.2 | 17.8 | 24.8 | 26.2 |
| Full | 30.4 | 69.3 | 37.6 | 65.8 | 50.2 | 87.8 | 21.5 | 51.8 | 76.2 | 85.2 | 66.8 | 63.0 | 72.8 | 33.5 | 56.8 | 40.3 | 66.8 | 73.5 | 40.4 | 28.0 | 22.8 | 45.3 | 53.5 |
| Linear | 41.4 | 83.9 | 62.0 | 76.5 | 88.8 | 37.6 | 35.9 | 60.9 | 80.5 | 88.6 | 69.4 | 74.0 | 78.1 | 35.3 | 34.0 | 34.0 | 65.7 | 18.3 | 20.5 | 15.3 | 21.5 | 30.6 | 51.8 |
| Fact-TK | 54.4 | 86.6 | 65.4 | 89.3 | 89.8 | 85.1 | 39.6 | 72.9 | 84.4 | 93.4 | 79.9 | 71.7 | 82.3 | 71.9 | 58.0 | 44.0 | 74.5 | 78.8 | 50.6 | 25.2 | 29.7 | 54.1 | 67.0 |
| LoRA | 55.3 | 87.6 | 67.3 | 90.2 | 90.2 | 85.1 | 39.6 | 73.6 | 81.7 | 94.1 | 80.5 | 72.7 | 82.2 | 77.6 | 59.1 | 47.0 | 81.6 | 81.5 | 49.6 | 29.4 | 35.1 | 57.6 | 68.7 |
| Adaptformer | 56.0 | 88.0 | 65.6 | 90.5 | 90.5 | 85.3 | 39.5 | 73.6 | 84.0 | 93.8 | 82.2 | 72.7 | 83.2 | 77.5 | 59.0 | 46.7 | 79.3 | 83.7 | 50.4 | 30.5 | 35.4 | 57.8 | 69.0 |
| SPT-LoRA | 55.8 | 89.5 | 68.1 | 90.6 | 88.4 | 40.6 | 90.2 | 74.8 | 84.5 | 95.7 | 84.8 | 74.1 | 84.8 | 73.8 | 58.5 | 47.4 | 80.5 | 80.4 | 50.0 | 29.8 | 35.7 | 57.0 | 69.4 |
| Adapter+ | 56.6 | 89.2 | 66.8 | 91.1 | 90.2 | 88.2 | 40.7 | 74.7 | 84.8 | 94.2 | 82.9 | 72.4 | 83.6 | 76.8 | 59.7 | 48.4 | 80.5 | 87.8 | 51.9 | 32.4 | 33.1 | 58.8 | 69.9 |
| **(A) Vim-S (T) DeiT (D) IN1K** | | | | | | | | | | | | | | | | | | | | | | | |
| Scratch | 5.6 | 11.9 | 5.9 | 12.1 | 5.0 | 16.3 | 1.6 | 8.3 | 61.6 | 62.3 | 13.8 | 61.6 | 49.8 | 28.9 | 53.2 | 22.5 | 40.9 | 38.6 | 11.8 | 11.3 | 18.3 | 28.2 | 25.4 |
| Full | 51.6 | 83.0 | 61.5 | 71.1 | 45.0 | 88.4 | 14.9 | 59.4 | 66.5 | 88.1 | 63.3 | 57.1 | 68.7 | 48.8 | 60.4 | 38.8 | 55.0 | 6.3 | 11.8 | 30.0 | 24.1 | 34.4 | 50.8 |
| Linear | 48.3 | 85.2 | 59.5 | 75.5 | 88.7 | 40.7 | 39.6 | 62.5 | 78.8 | 89.0 | 68.2 | 73.0 | 77.3 | 37.0 | 36.7 | 33.3 | 66.0 | 18.0 | 25.2 | 16.5 | 23.2 | 32.0 | 52.8 |
| CLS-token-tuning | 48.0 | 85.1 | 59.2 | 75.7 | 88.8 | 41.0 | 39.4 | 62.4 | 78.7 | 89.2 | 68.5 | 73.1 | 77.4 | 37.3 | 36.7 | 33.6 | 66.4 | 18.3 | 25.4 | 17.1 | 23.5 | 32.3 | 52.9 |
| Pos-embed-tuning | 50.6 | 85.1 | 59.4 | 74.6 | 88.3 | 50.5 | 39.9 | 64.1 | 77.2 | 87.6 | 68.2 | 66.4 | 74.8 | 32.9 | 46.3 | 32.4 | 65.0 | 58.6 | 29.8 | 19.0 | 29.5 | 39.2 | 55.9 |
| D-tuning | 54.2 | 86.0 | 63.1 | 81.3 | 88.9 | 55.5 | 41.1 | 67.1 | 78.8 | 91.3 | 73.0 | 71.2 | 78.6 | 45.3 | 42.7 | 36.9 | 70.0 | 41.5 | 38.5 | 19.9 | 26.0 | 40.1 | 58.2 |
| A-tuning | 58.7 | 87.4 | 65.0 | 86.5 | 90.2 | 74.0 | 42.4 | 72.0 | 81.1 | 93.2 | 78.4 | 70.2 | 80.7 | 62.7 | 53.9 | 40.0 | 77.5 | 60.6 | 46.0 | 23.9 | 32.1 | 49.6 | 64.4 |
| Conv1d-tuning | 59.7 | 89.1 | 65.2 | 86.5 | 89.4 | 88.4 | 42.1 | 74.3 | 82.9 | 94.2 | 79.2 | 73.5 | 82.5 | 76.5 | 63.3 | 45.7 | 79.5 | 80.2 | 51.3 | 30.8 | 35.4 | 57.8 | 69.1 |
| Prompt-tuning | 55.6 | 87.2 | 62.7 | 82.8 | 89.5 | 70.3 | 41.5 | 69.9 | 78.1 | 91.4 | 74.9 | 72.4 | 79.2 | 59.1 | 54.8 | 38.8 | 73.7 | 58.9 | 45.7 | 22.1 | 28.9 | 47.8 | 62.5 |
| Affix-tuning | 61.9 | 89.3 | 67.8 | 89.7 | 91.0 | 88.0 | 43.2 | 75.8 | 82.6 | 94.8 | 82.2 | 73.6 | 83.3 | 80.4 | 59.5 | 46.1 | 81.3 | 83.6 | 50.9 | 30.5 | 39.4 | 58.9 | 70.3 |
| Additional-scan | 59.6 | 90.0 | 66.6 | 88.9 | 90.7 | 83.6 | 43.1 | 74.6 | 83.2 | 94.4 | 81.3 | 71.9 | 82.7 | 75.1 | 63.2 | 43.7 | 80.0 | 75.0 | 48.1 | 29.1 | 36.9 | 56.4 | 68.7 |
| ParallelAdapter | 64.0 | 89.7 | 68.9 | 91.4 | 91.3 | 83.4 | 44.1 | 76.1 | 85.3 | 95.2 | 83.4 | 72.0 | 84.0 | 83.9 | 63.4 | 49.8 | 80.2 | 75.5 | 53.0 | 32.8 | 41.3 | 60.0 | 71.0 |
| LoRA(embed) | 51.7 | 84.9 | 59.2 | 75.5 | 88.0 | 53.4 | 39.8 | 64.7 | 77.5 | 91.2 | 71.5 | 70.0 | 77.5 | 42.9 | 52.0 | 34.9 | 73.0 | 54.9 | 47.0 | 19.7 | 26.2 | 43.8 | 58.6 |
| LoRA(x_proj) | 59.2 | 88.0 | 67.5 | 88.9 | 90.4 | 83.9 | 43.0 | 74.4 | 82.7 | 93.9 | 80.3 | 70.7 | 81.9 | 72.5 | 60.7 | 43.6 | 80.6 | 75.0 | 45.8 | 27.8 | 33.1 | 54.9 | 67.8 |
| LoRA(dt_proj) | 61.6 | 90.1 | 66.7 | 89.0 | 90.7 | 86.1 | 43.4 | 75.4 | 83.2 | 94.9 | 81.2 | 72.9 | 83.1 | 76.9 | 59.8 | 47.0 | 79.8 | 75.6 | 50.7 | 30.3 | 37.0 | 57.1 | 69.3 |
| LoRA(out_proj) | 64.2 | 89.8 | 68.6 | 91.4 | 91.2 | 86.0 | 43.8 | 76.4 | 84.9 | 94.9 | 83.4 | 73.2 | 84.1 | 84.5 | 62.7 | 49.5 | 81.3 | 77.1 | 52.0 | 32.8 | 40.9 | 60.1 | 71.1 |
| LoRA(in_proj) | 63.4 | 90.3 | 68.2 | 91.0 | 91.0 | 88.6 | 43.5 | 76.6 | 84.7 | 95.0 | 83.8 | 72.9 | 84.1 | 84.0 | 61.3 | 48.6 | 80.0 | 83.5 | 52.6 | 31.8 | 39.5 | 60.2 | 71.3 |
| LoRA$_p$(d) | 57.1 | 87.4 | 65.9 | 87.6 | 90.4 | 81.8 | 42.7 | 73.3 | 80.9 | 93.6 | 78.9 | 70.2 | 80.9 | 64.2 | 55.4 | 42.8 | 76.9 | 64.5 | 47.4 | 25.5 | 30.8 | 50.9 | 65.5 |
| LoRA$_p$(C) | 55.8 | 87.8 | 66.9 | 87.6 | 90.5 | 78.6 | 42.3 | 72.8 | 82.1 | 94.1 | 79.7 | 70.5 | 81.6 | 60.5 | 59.2 | 41.3 | 80.9 | 70.4 | 42.8 | 25.9 | 29.7 | 51.4 | 65.6 |
| LoRA$_p$(B) | 56.5 | 87.5 | 66.3 | 88.1 | 90.5 | 79.0 | 42.8 | 73.0 | 82.9 | 93.9 | 79.6 | 70.3 | 81.7 | 67.2 | 58.8 | 41.5 | 78.3 | 69.4 | 44.4 | 26.8 | 31.7 | 52.3 | 66.1 |
| LoRA$_p$(Z) | 61.7 | 90.6 | 68.1 | 90.3 | 90.7 | 88.4 | 43.2 | 76.2 | 85.6 | 95.2 | 83.5 | 72.8 | 84.3 | 82.4 | 60.6 | 48.2 | 81.9 | 81.6 | 52.4 | 31.4 | 39.5 | 59.7 | 70.9 |
| LoRA$_p$(X) | 63.3 | 89.9 | 69.4 | 90.6 | 91.5 | 88.9 | 43.6 | 76.6 | 84.9 | 95.3 | 83.4 | 72.4 | 83.9 | 84.9 | 62.9 | 48.6 | 81.4 | 82.8 | 52.7 | 33.1 | 40.4 | 60.8 | 71.5 |
| Hybrid | 63.7 | 90.2 | 69.4 | 90.9 | 90.1 | 90.1 | 43.9 | 77.0 | 86.3 | 95.2 | 83.9 | 72.2 | 84.4 | 84.6 | 61.7 | 50.0 | 81.0 | 86.4 | 54.0 | 34.6 | 40.2 | 61.6 | 72.1 |
| **(A) ViT-tiny (T) DeiT (D) IN1K** | | | | | | | | | | | | | | | | | | | | | | | |
| Linear | 35.7 | 81.3 | 57.1 | 72.4 | 84.5 | 35.2 | 29.9 | 56.6 | 81.0 | 86.8 | 67.0 | 74.2 | 77.2 | 32.5 | 32.7 | 33.1 | 59.9 | 14.2 | 17.7 | 13.6 | 19.2 | 27.9 | 48.8 |
| FacT-TK | 44.7 | 83.6 | 59.5 | 85.9 | 91.5 | 81.6 | 32.9 | 68.5 | 82.0 | 91.4 | 75.0 | 70.0 | 79.6 | 66.3 | 58.3 | 43.1 | 72.9 | 77.4 | 45.0 | 24.3 | 31.1 | 52.3 | 64.0 |
| LoRA | 45.5 | 85.9 | 59.5 | 85.3 | 86.2 | 84.7 | 32.3 | 68.5 | 80.5 | 93.0 | 76.1 | 70.9 | 80.1 | 72.2 | 60.0 | 45.4 | 76.1 | 79.7 | 45.9 | 27.6 | 32.8 | 55.0 | 65.2 |
| SPT-LoRA | 47.3 | 85.0 | 62.8 | 86.3 | 86.3 | 85.6 | 34.4 | 69.7 | 83.3 | 94.5 | 80.6 | 72.8 | 82.8 | 64.6 | 58.6 | 47.4 | 79.5 | 79.7 | 48.6 | 30.9 | 33.6 | 55.4 | 66.4 |
| Adapter+ | 47.5 | 86.4 | 61.2 | 92.9 | 87.2 | 78.9 | 33.9 | 69.7 | 85.9 | 77.9 | 83.0 | 71.6 | 79.6 | 72.7 | 58.5 | 46.3 | 78.0 | 83.0 | 47.1 | 29.4 | 34.2 | 56.2 | 66.1 |
| **(A) Vim-tiny (T) DeiT (D) IN1K** | | | | | | | | | | | | | | | | | | | | | | | |
| Linear | 43.0 | 83.7 | 56.9 | 73.7 | 87.4 | 40.6 | 36.9 | 60.3 | 79.0 | 87.1 | 68.3 | 73.1 | 76.9 | 32.6 | 36.1 | 34.2 | 68.5 | 22.4 | 15.0 | 21.3 | 30.4 |  | 51.2 |
| Additional-scan | 52.6 | 87.1 | 62.3 | 85.7 | 88.5 | 81.7 | 37.0 | 70.7 | 80.2 | 93.0 | 78.8 | 72.0 | 81.0 | 69.6 | 62.2 | 42.6 | 78.6 | 77.0 | 43.5 | 27.0 | 32.4 | 54.1 | 65.9 |
| Conv1d-tuning | 50.0 | 86.4 | 60.0 | 82.8 | 85.6 | 89.2 | 35.2 | 69.8 | 82.5 | 93.3 | 77.7 | 69.7 | 80.8 | 71.5 | 63.9 | 44.1 | 77.1 | 77.9 | 48.1 | 29.9 | 34.4 | 55.9 | 66.3 |
| Affix-tuning | 53.3 | 87.4 | 60.4 | 85.9 | 88.4 | 84.4 | 37.3 | 66.8 | 79.4 | 93.8 | 77.7 | 70.0 | 80.2 | 74.3 | 64.2 | 44.2 | 77.9 | 79.6 | 48.0 | 27.0 | 38.3 | 56.7 | 66.9 |
| LoRA$_p$(X) | 56.0 | 87.6 | 64.5 | 87.7 | 89.1 | 87.4 | 38.2 | 72.9 | 83.2 | 94.5 | 82.2 | 71.8 | 80.8 | 78.7 | 59.3 | 47.2 | 76.1 | 81.0 | 50.6 | 29.7 | 37.4 | 57.5 | 68.5 |

## A.4 ABLATION STUDIES ON LANGUAGE TASKS

We conduct ablation studies for language tasks. Figure 6c shows that the optimal learning rate varies for each method. In contrast, as shown in Figure 6c and Figure 6d, the optimal learning rate and the optimal number of trainable parameters remain relatively stable despite changes in model size. Hence, the number of trainable parameters is fixed regardless of the model size, and the learning rate is coarsely tuned for evaluation in Table 3. The detailed settings are described in Appendix B.1.

The trade-off between the number of trainable parameters and accuracy for Additional-scan, Affix-tuning (w/o proj), and LoRA$_p$(X) is described in Figure 6b. Each method excels in certain aspects

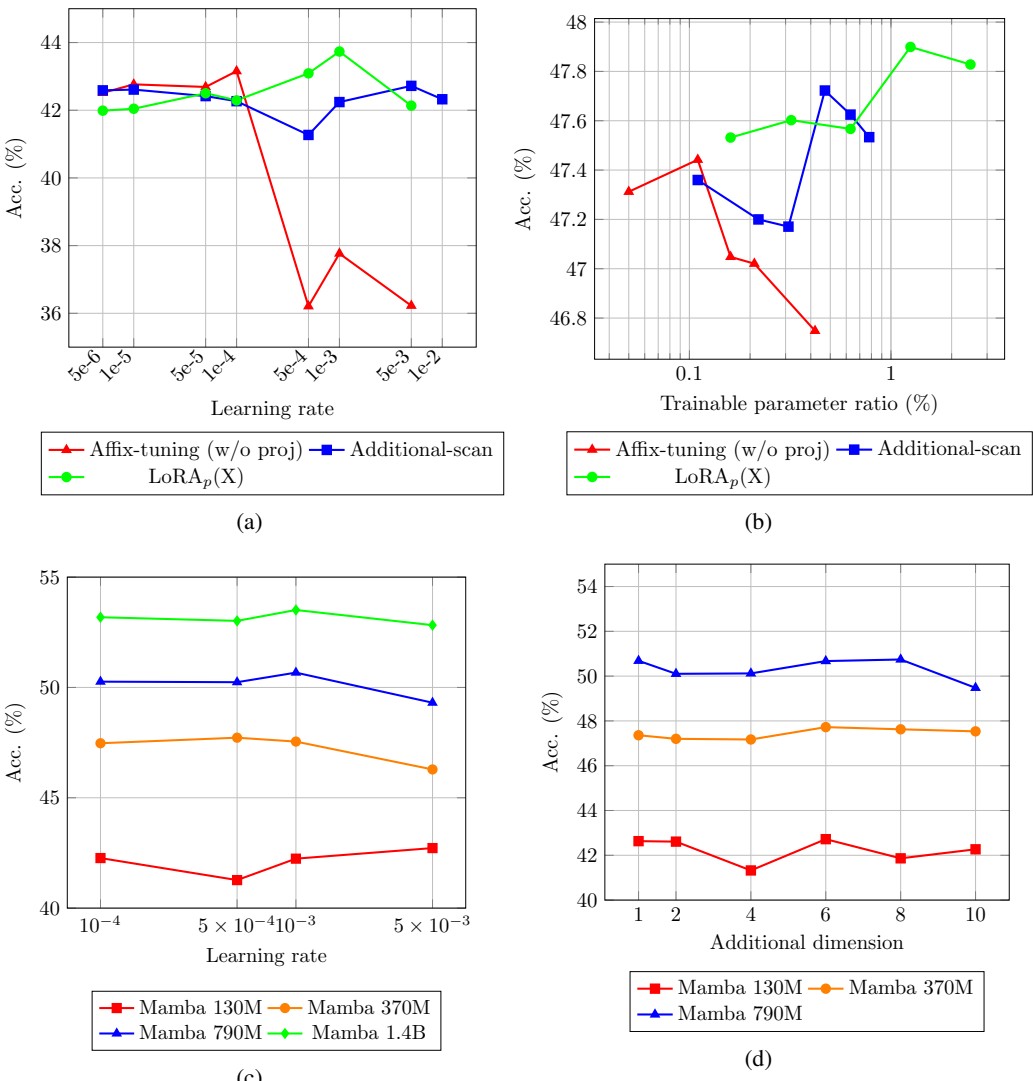

Figure 6: This ablation study investigates how optimal hyperparameters differ per PEFT method and the size of base models in language tasks. **(a)** Optimal learning rate per PEFT method. **(b)** The relationship between the ratio of trainable parameters and performance on Mamba 370M base model. **(c)** Suitable learning rate per base model size for Additional-scan. **(d)** Optimal additional dimension per base model size for Additional-scan.

and should be chosen based on the specific application. LoRA$_p$(X) is effective when sufficient memory is available, Affix-tuning (w/o proj) maintains accuracy with the fewest trainable parameters, and Additional-scan falls in between. Since even PEFT may not fit into GPU memory with large-scale models, it is crucial to select the appropriate method based on the application and environment.

Table 6 shows the experimental results of applying LoRA to individual modules in both Mamba and Pythia. In Mamba, applying LoRA to individual modules yielded better performance than SLL LoRA, which applies LoRA to all modules. Conversely, in Pythia, applying LoRA to all modules tended to perform better than applying LoRA to individual modules. These experimental results demonstrate the differences in behavior between Mamba and Transformer when LoRA is applied.

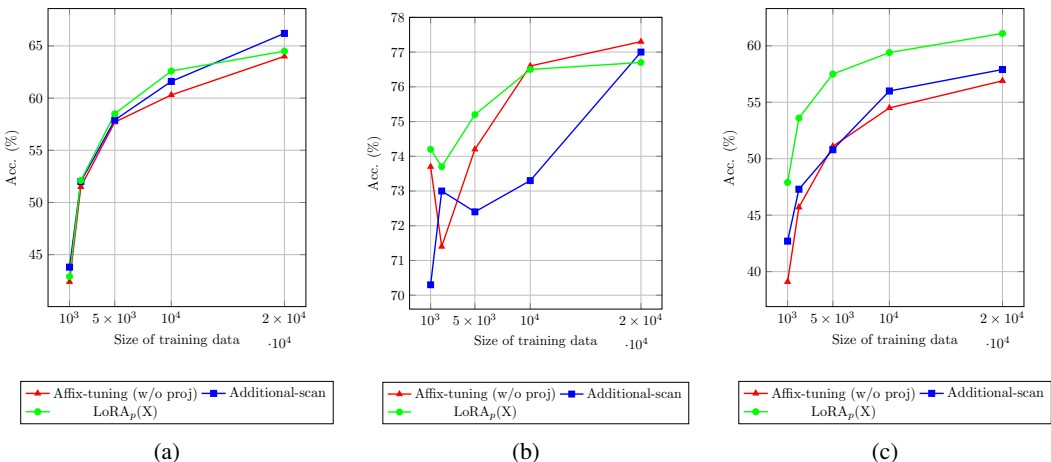

Figure 7: Task-specific relationship between size of training data and performance of PEFT in Vim. (a) is on Sun397, (b) is on Retinopathy, and (c) is on DMLab task.

| Base Model | Method | #Params (K) | Natural | Specialized | Structured | Average |
|---|---|---|---|---|---|---|
| VMamba-S | Full | 49,379 | 66.4 | 81.8 | 62.6 | 68.0 |
| | Linear | 18 | 67.9 | 73.6 | 34.2 | 54.9 |
| | Conv2d-tuning | 160 | 75.8 | 83.3 | 58.9 | 70.3 |
| | Affix-tuning(w/o proj) | 198 | 77.3 | 82.7 | 59.7 | 71.0 |
| | Additional-scan | 395 | 76.4 | 83.3 | 60.2 | 71.0 |
| | ParallelAdapter | 774 | 77.8 | 84.5 | 63.7 | 73.2 |
| | LoRA(out_proj) | 1,529 | 77.9 | **85.9** | **65.6** | **74.4** |
| | LoRA$_p$(X) | 1,529 | **78.2** | 84.5 | 65.2 | 74.0 |
| VMamba-B | Full | 87,533 | 66.8 | 81.4 | 60.1 | 67.1 |
| | Linear | 24 | 68.4 | 79.1 | 30.2 | 54.6 |
| | Conv2d-tuning | 189 | 78.0 | 84.1 | 63.4 | 73.1 |
| | Affix-tuning(w/o proj) | 276 | 77.5 | 83.3 | 59.2 | 71.0 |
| | Additional-scan | 527 | 77.0 | 83.8 | 60.1 | 71.3 |
| | ParallelAdapter | 1,032 | **78.6** | 85.1 | 63.3 | 73.5 |
| | LoRA(out_proj) | 2,039 | **78.6** | **85.7** | 64.9 | **74.3** |
| | LoRA$_p$(X) | 2,039 | 78.3 | 85.3 | **65.2** | **74.3** |

Table 5: Comparison of methods for VMamba. In VMamba, LoRA$_p$(X) is the same as LoRA(in_proj).

## A.5 EFFECT OF DATA SIZE

Table 1 demonstrates that Partial LoRA performs better in image tasks. On the other hand, Table 3 shows that Affix-tuning and Additional-scan achieve competitive accuracy with Partial LoRA despite fewer parameters in language tasks. The major differences between the two experiments are the modality and the amount of training data. In the image experiment, fine-tuning was performed with 1K images, while in the language experiment, fine-tuning was conducted with 170K data. To investigate whether the differences in trends are due to the amount of data, we conducted experiments to scale the amount of training data in the image tasks. In detail, we used official test data of VTAB-1K as training data and vice versa. We chose Sun397, Retinopathy, and DMLab from each category of VTAB-1K, which contains a large number of images for this experiment. fig. 7 shows that, as the data size increases, the best methods become Additional-scan for (a), Affix-tuning for (b), and LoRA$_p$(X) for (c). These experiments suggest that LoRA$_p$(X) excels with smaller data sizes, but with larger data sizes, the optimal method depends on the specific task. We suggest selecting a method that balances the trade-off between performance and parameters based on the task or searching HybridPEFT.

Table 6: Experimental results applying LoRA to individual modules in Pythia and Mamba.

| Model | Method | BoolQ | PIQA | SIQA | HellaSwag | WinoGrande | ARC-e | ARC-c | OBQA | Avg. |
|---|---|---|---|---|---|---|---|---|---|---|
| Pythia 160M | Full | 61.3 | 62.9 | 37.1 | 30.7 | 50.6 | 41.5 | 24.3 | 27.8 | 42.0 |
| | LoRA | 61.0 | 62.0 | 36.3 | 30.3 | 52.0 | 38.2 | 24.6 | 28.0 | 41.6 |
| | LoRA(query_key_value) | 50.4 | 62.0 | 36.8 | 30.2 | 50.7 | 40.5 | 23.9 | 27.0 | 40.2 |
| | LoRA(dense_4h_to_h) | 50.7 | 61.7 | 37.9 | 29.9 | 49.6 | 39.7 | 22.4 | 27.0 | 39.8 |
| | LoRA(dense_h_to_4h) | 55.5 | 61.3 | 37.9 | 29.9 | 48.7 | 36.2 | 22.8 | 26.8 | 39.9 |
| | LoRA(dense) | 49.9 | 61.6 | 36.9 | 30.1 | 51.1 | 41.1 | 23.3 | 26.0 | 40.0 |
| Mamba 130M | Full | 56.1 | 65.3 | 38.7 | 35.3 | 52.0 | 46.4 | 25.7 | 32.8 | 43.8 |
| | SLL LoRA | 56.3 | 63.3 | 38.2 | 34.6 | 51.6 | 43.5 | 23.6 | 30.6 | 42.7 |
| | Affix-tuning (w/o proj) | 55.1 | 61.4 | 36.5 | 32.9 | 51.5 | 36.8 | 23.5 | 27.2 | 40.6 |
| | LoRA(in_proj) | 53.5 | 62.9 | 38.2 | 33.8 | 53.1 | 46.4 | 23.7 | 30.8 | 42.8 |
| | LoRA$_p$(X) | 61.7 | 64.0 | 39.5 | 34.3 | 52.2 | 43.5 | 25.3 | 29.4 | 43.7 |
| Pythia 410M | Full | 55.0 | 68.4 | 42.1 | 40.8 | 53.9 | 50.8 | 26.7 | 30.0 | 46.0 |
| | LoRA | 61.3 | 67.7 | 40.8 | 39.2 | 54.9 | 48.1 | 24.7 | 28.6 | 45.7 |
| | LoRA(query_key_value) | 60.5 | 67.5 | 41.3 | 39.2 | 54.3 | 45.4 | 24.0 | 29.0 | 45.1 |
| | LoRA(dense_4h_to_h) | 61.2 | 67.1 | 40.5 | 39.0 | 54.5 | 47.1 | 25.1 | 28.6 | 45.4 |
| | LoRA(dense_h_to_4h) | 61.5 | 67.0 | 39.9 | 39.0 | 53.7 | 46.2 | 23.8 | 28.6 | 45.0 |
| | LoRA(dense) | 55.7 | 67.4 | 40.5 | 39.8 | 54.3 | 45.2 | 24.1 | 30.6 | 44.7 |
| Mamba 370M | Full | 58.1 | 69.9 | 41.9 | 45.7 | 53.8 | 52.7 | 29.7 | 33.4 | 48.2 |
| | SLL LoRA | 59.5 | 69.6 | 42.2 | 44.1 | 54.9 | 50.6 | 26.3 | 30.8 | 47.3 |
| | Affix-tuning (w/o proj) | 62.0 | 67.7 | 39.3 | 46.3 | 54.1 | 47.8 | 28.2 | 31.0 | 47.0 |
| | LoRA(in_proj) | 55.4 | 68.6 | 41.0 | 44.7 | 54.1 | 52.4 | 28.3 | 33.4 | 47.2 |
| | LoRA$_p$(X) | 60.8 | 68.8 | 42.1 | 44.7 | 56.2 | 50.4 | 27.4 | 32.2 | 47.8 |
| Pythia 1B | Full | 55.0 | 70.2 | 42.5 | 47.5 | 54.4 | 54.1 | 29.7 | 33.2 | 48.3 |
| | LoRA | 60.0 | 69.3 | 40.9 | 45.3 | 53.6 | 49.8 | 27.2 | 31.0 | 47.1 |
| | LoRA(query_key_value) | 58.0 | 69.4 | 41.7 | 44.7 | 54.1 | 50.3 | 27.3 | 30.8 | 47.0 |
| | LoRA(dense_4h_to_h) | 49.2 | 69.0 | 42.0 | 45.2 | 53.8 | 52.3 | 27.5 | 31.0 | 46.3 |
| | LoRA(dense_h_to_4h) | 53.1 | 68.9 | 38.9 | 45.0 | 52.2 | 49.7 | 26.5 | 30.2 | 45.6 |
| | LoRA(dense) | 53.3 | 69.4 | 41.0 | 46.5 | 52.4 | 49.4 | 27.8 | 32.6 | 46.6 |
| Mamba 790M | Full | 62.0 | 72.1 | 44.8 | 54.0 | 55.9 | 57.7 | 31.2 | 35.2 | 51.6 |
| | SLL LoRA | 60.7 | 72.0 | 42.4 | 54.7 | 56.9 | 55.3 | 29.4 | 34.2 | 50.7 |
| | Affix-tuning (w/o proj) | 57.2 | 71.7 | 41.4 | 55.0 | 55.8 | 52.6 | 29.8 | 33.0 | 49.6 |
| | LoRA(in_proj) | 61.7 | 71.9 | 44.0 | 50.8 | 56.7 | 56.3 | 30.5 | 33.8 | 50.7 |
| | LoRA$_p$(X) | 59.9 | 72.2 | 44.2 | 52.8 | 58.0 | 53.7 | 30.8 | 34.8 | 50.8 |
| Pythia 1.4B | Full | 58.6 | 71.1 | 42.7 | 53.6 | 55.1 | 58.5 | 29.9 | 34.8 | 50.5 |
| | LoRA | 60.1 | 71.3 | 42.5 | 50.1 | 58.9 | 57.6 | 29.6 | 33.6 | 50.5 |
| | LoRA(query_key_value) | 60.7 | 70.9 | 42.2 | 50.0 | 58.3 | 55.4 | 28.2 | 33.2 | 49.9 |
| | LoRA(dense_4h_to_h) | 62.2 | 71.9 | 41.2 | 50.5 | 57.6 | 55.6 | 28.8 | 33.6 | 50.2 |
| | LoRA(dense_h_to_4h) | 61.4 | 70.7 | 42.9 | 50.3 | 57.6 | 51.7 | 26.7 | 32.8 | 49.3 |
| | LoRA(dense) | 62.1 | 71.1 | 41.6 | 51.6 | 58.2 | 54.7 | 29.4 | 31.6 | 50.0 |
| Mamba 1.4B | Full | 61.4 | 73.3 | 43.9 | 56.9 | 59.0 | 59.7 | 34.0 | 35.4 | 53.0 |
| | SLL LoRA | 59.7 | 73.5 | 43.1 | 56.9 | 60.7 | 59.7 | 31.7 | 36.0 | 52.7 |
| | Affix-tuning (w/o proj) | 63.6 | 74.0 | 41.9 | 58.9 | 60.6 | 61.6 | 33.4 | 36.8 | 53.9 |
| | LoRA(in_proj) | 62.6 | 73.6 | 43.7 | 55.6 | 59.7 | 58.3 | 31.7 | 35.6 | 52.6 |
| | LoRA$_p$(X) | 63.1 | 73.5 | 42.7 | 57.7 | 61.6 | 60.4 | 32.9 | 37.4 | 53.7 |

## A.6 EFFECT OF MODEL SIZE

Table 4 contains experimental results for Vim and ViT with size variations. From Vim-Tiny to Vim-S, the overall accuracy is improved. However, from Vim-S to Vim-B, there is little improvement, whereas ViT-B improved from ViT-S. Because this phenomenon is not observed in vanilla Mamba for language tasks (see table 3), it seems it is not due to PEFT methods for Mamba. This suggests that large Vim/Mamba models might face potential challenges when fine-tuning with limited data. Note that even with Vim-B, PEFT still significantly outperforms full fine-tuning and achieves equivalent or higher accuracy than state-of-the-art PEFT with ViT-B, indicating that PEFT for Vim remains effective.

## A.7 EXPERIMENTS WITH VMAMBA

To investigate the generality of PEFT in Mamba for visual tasks, we conduct experiments using VMamba (Liu et al., 2024b), a variant other than Vim. VMamba made several improvements over the original Mamba. It eliminated the state dimension (i.e., reduced the state dimension to 1), introduced a hierarchical structure similar to CNNs, replaced causal 1D convolutions with 2D convolutions, removed the Gated MLP structure, and substituted half of the Mamba blocks with MLP blocks. Due to these changes, there are some differences in trends, but PEFT also works well with VMamba, as shown in Table 5. The accuracy is higher than that of Vim, as the original VMamba performs better on ImageNet1K evaluation. Similar to the results with Vim, partial LoRA shows

Table 7: Replacing simple structure PEFT methods we intentionally used with dedicated PEFT methods.

| Method | VTAB-1K with Vim-S | | | | Commonsense reasoning with Mamba 1.4B | | | | | | | | |
| | Natu. | Spec. | Struc. | Avg. | BoolQ | PIQA | SIQA | HellaS | WinoG | ARC-e | ARC-c | OBQA | Avg. |
|---|---|---|---|---|---|---|---|---|---|---|---|---|---|
| LoRA$_p$(X) | 76.6 | 83.9 | **60.8** | 71.5 | 63.1 | 73.5 | 42.7 | 57.7 | 61.6 | **60.4** | **32.9** | **37.4** | 53.7 |
| DoRA$_p$(X) | **76.9** | **84.1** | 60.6 | **71.6** | **63.9** | **73.8** | **43.2** | **57.9** | **62.1** | **60.4** | 32.4 | **37.4** | **53.9** |

superior performance. Additionl-scan and Partial-tuning show a superior trade-off between trainable parameters and performance. Furthermore, there was no significant performance improvement from VMamba-S to VMamba-B, indicating saturation, which is also consistent with Vim. These experimental results suggest that the phenomenon of performance saturation with PEFT in Mamba for visual tasks, once the model size reaches a certain threshold, is general regardless of variation.

## A.8    COMPARISON WITH CONCURRENT WORK

We discuss the comparison with related studies conducted during the same period in addition to SLL LoRA (Halloran et al., 2024). Galim et al. (2024) proposed SDLoRA, which selectively updates the channels and states of SSM. Unlike SDLoRA, our work emphasizes comprehensive benchmarking and exploration. Furthermore, Galim et al. (2024) pointed out the limitations of expressiveness when applying Prefix-tuning to SSM. In contrast, our proposed Affix-tuning avoids this issue by allowing tokens to be inserted at arbitrary positions. In fact, Affix-tuning shows competitive performance with LoRA on language tasks (table 3).

## A.9    DIRECTION OF THE FUTURE WORK

Our goal is to explore PEFT for Mamba: specifically, how to tune and where to tune. By using a simple architecture PEFT methods, we aim to minimize extraneous influences, allowing evaluation results to be universally informative for various future tuning strategies. The proposed Affix-tuning and Additional-scan are also designed as simple as possible, although we believe they possess significant novelty and usefulness. Based on our findings, future work should be able to focus on effective PEFT research without searching a lot. To verify this, we conducted an experiment by replacing LoRA with the latest DoRA (Liu et al., 2024a). As shown in Table 7, we were able to achieve even higher accuracy than LoRA$_p$(X), which already surpasses the accuracy of state-of-the-art dedicated PEFT for Transformers. We hope that our exploration will contribute to the future evolution of PEFT for Mamba.

## B    IMPLEMENTATION DETAILS

### B.1    LANGUAGE TASKS

We follow the fine-tuning setup of Liu et al. (2024a); Hu et al. (2023) for commonsense reasoning tasks. Each model is fine-tuned with about 140,000 data for three epochs with a batch size of 16. A linear learning rate scheduler is used with a warmup period of 100 iterations.

As to the learning rate, we use suitable values for each method. For SLL LoRA on Mamba, we follow the original learning rate per model size (Halloran et al., 2024). We also use a roughly tuned learning rate for our PEFT methods obtained through the ablation studies in Appendix A.4. The configurations are shown in Table 8.

Table 8: The learning rate configurations for language tasks.

| Method | Mamba 130M | Mamba 370M | Mamba 790M | Mamba 1.4B |
|---|---|---|---|---|
| Full | 5e-5 | 5e-5 | 5e-5 | 5e-5 |
| Additional-scan | 5e-3 | 1e-3 | 1e-3 | 1e-4 |
| Affix-tuning | 1e-4 | 1e-5 | 1e-5 | 1e-5 |
| LoRA$_p$(X) | 1e-3 | 5e-4 | 5e-4 | 1e-4 |

| Method | Pythia 160M | Pythia 410M | Pythia 1B | Pythia 1.4B |
|---|---|---|---|---|
| Full | 5e-5 | 5e-5 | 5e-5 | 5e-6 |
| LoRA | 1e-5 | 1e-4 | 1e-5 | 1e-5 |

For the hyperparameters of each PEFT method, we use the optimal values obtained through the ablation studies on the VTAB-1K dataset. For Pythia (Biderman et al., 2023), We apply LoRA with $r = 8$ to dense, query_key_value, dense_4h_to_h, and dense_h_to_4h layers.

## B.2 HYBRID PEFT SEARCH

In the first step, we explore only whether to use each PEFT method. As mentioned in the Section 3.4, the first step of our search verifies the effectiveness of each PEFT method and seeks the preferable combination with minimum trainable parameters. We rely on the following hypothesis; effective PEFT methods work positively even when the number of trainable parameters is small, while when the number of parameters increases beyond a certain level, the pre-trained model collapses and is negatively affected. As table 9 shows, simply combining PEFT or increasing the rank of LoRA decreases performance. Therefore, the hyperparameters of each PEFT method are set as its trainable parameters become minimum. In addition, the proposed regularization for Partial-tuning PEFT methods is applied so as not to corrupt the pre-trained model. In this step, a combination of PEFT methods that work positively is selected greedy while keeping the total degrees of freedom of the PEFT methods within the degrees of freedom that do not destroy the pre-trained model. We search for 100 trials using TPE algorithm (Bergstra et al., 2011) about whether to use each PEFT method with the fixed hyperparameters shown in step 1 of Table 10. As a result, it turns out that the combination of eight PEFT methods, CLS-token-tuning, A-tuning, Affix-tuning, LoRA(out_proj), LoRA(in_proj), LoRA(dt_proj), LoRA$_p$(X), and LoRA$_p$(B), is the best.

In the second step, the hyperparameters for the eight methods are searched. In addition, a search dimension is added, which selects a PEFT method that should be removed or decides not to remove any at all. This is intended to eliminate less important PEFT methods and allocate more parameters to the more significant ones. The specific search space is shown in step 2 of Table 10.

This two-step approach reduces the search space to only $2^{N_{\text{all}}}$ for the first step and $(N_i+1) \prod_{p \in P_i} d_p$ for the subsequent steps, compared to a simultaneous search space with $2^{N_{\text{all}}} \prod_{p \in P_{\text{all}}} d_p$. $N_{\text{all}}$ is the number of PEFT methods, and $N_i (\leq N_{\text{all}})$ is the number of methods chosen in the previous search step. The $P_{\text{all}}$ is the set of all the hyperparameters and $P_i (\subseteq P_{\text{all}})$ is the subset used in the chosen methods. Also, we represent the search space size along a parameter $p$ as $d_p$.

Since optimizing over all 19 tasks is computationally expensive, we optimize over the average accuracy of 5 tasks in the representative subset of VTAB-1k (Zhai et al., 2019): Caltech101, EuroSAT, Pets, Camelyon, and Resisc45. By processing five tasks in parallel on one A100 GPU, one trial can be completed in around 20 minutes, with minimal dependency on the type and size of the applied PEFT methods.

The detailed algorithm is provided in Algorithm 1. In our experiments, we use TPE as the search algorithm, top-1 accuracy as the objective function, and the number of trials is set to $N = 100$.

---

**Algorithm 1** HybridPEFT Search Algorithm

---

1: **Input:** Set of PEFT methods $\mathcal{M}$, Hyperparameter space $\mathcal{H}$, Training hyperparameter space $\mathcal{T}$, Default hyperparameters $(h_{\text{default}}, t_{\text{default}})$, Objective function $f$, Number of iterations $N$, Search algorithm $\mathcal{S}$
2: **Output:** Optimal combination of PEFT methods and hyperparameters $\mathcal{C}^*, h^*, t^*$
3: **Step 1: Search for PEFT Method Combinations using $\mathcal{S}$**
4: Initialize set of active PEFT methods $\mathcal{C} \leftarrow \emptyset$
5: Initialize set of observations $\mathcal{O}_{\text{combi}} \leftarrow \emptyset$
6: **for** iteration $i = 1$ to $N$ **do**
7:     Sample PEFT method combination $\mathcal{C}_i$ using $\mathcal{S}$ based on $\mathcal{O}_{\text{combi}}$
8:     Evaluate performance $f(\mathcal{C}_i, h_{\text{default}}, t_{\text{default}})$
9:     Add $(\mathcal{C}_i, f(\mathcal{C}_i, h_{\text{default}}, t_{\text{default}}))$ to $\mathcal{O}_{\text{combi}}$
10:     **if** performance improves **then**
11:         Update $\mathcal{C} \leftarrow \mathcal{C}_i$
12:     **end if**
13: **end for**
14: **Step 2: Search for Hyperparameters using $\mathcal{S}$**
15: Initialize set of active PEFT methods $\mathcal{C}^* \leftarrow \mathcal{C}$
16: Initialize best hyperparameters $h^* \leftarrow \emptyset, t^* \leftarrow \emptyset$
17: Initialize set of observations $\mathcal{O}_{\text{hyper}} \leftarrow \emptyset$
18: **for** iteration $j = 1$ to $N$ **do**
19:     **if** $j == 1$ **then**
20:         Use default hyperparameters $(h_j, t_j) \leftarrow (h_{\text{default}}, t_{\text{default}})$
21:         Initialize $r \leftarrow$ "not_remove"
22:     **else**
23:         Sample hyperparameters $(h_j, t_j)$ using $\mathcal{S}$ based on $\mathcal{O}_{\text{hyper}}$
24:         Sample a PEFT method to remove $r$ from $\{\mathcal{C} \cup$ "not_remove"$\}$ using $\mathcal{S}$ based on $\mathcal{O}_{\text{hyper}}$
25:     **end if**
26:     **if** $r$ is "not_remove" **then**
27:         Evaluate performance $f(\mathcal{C}, h_j, t_j)$
28:         Add $((h_j, t_j), f(\mathcal{C}, h_j, t_j))$ to $\mathcal{O}_{\text{hyper}}$
29:     **else**
30:         Evaluate performance $f(\mathcal{C} \setminus r, h_j, t_j)$
31:         Add $((h_j, t_j), f(\mathcal{C} \setminus r, h_j, t_j))$ to $\mathcal{O}_{\text{hyper}}$
32:     **end if**
33:     **if** performance improves **then**
34:         Update $h^* \leftarrow h_j, t^* \leftarrow t_j$
35:         **if** $r$ is "not_remove" **then**
36:             Update $\mathcal{C}^* \leftarrow \mathcal{C}$
37:         **else**
38:             Update $\mathcal{C}^* \leftarrow \mathcal{C} \setminus r$
39:         **end if**
40:     **end if**
41: **end for**
42: **Return** $\mathcal{C}^*, h^*, t^*$

---

| Method | Rank | Avg. |
|---|---|---|
| LoRA(in_proj + out_proj) | 8 | 71.33 |
| LoRA(in_proj + out_proj) | 16 | 70.68 |
| LoRA(in_proj + out_proj) | 32 | 69.77 |
| LoRA(ALL) | 8 | 71.02 |
| LoRA(ALL) | 16 | 70.42 |
| LoRA(ALL) | 32 | 68.74 |
| $LoRA_p(X)$ | 64 | **71.52** |

Table 9: Average performance of VTAB-1K when combining multiple LoRA.

Table 10: The search space and the fixed hyperparameters in our Hybrid PEFT search of the first step and the second step. In the first step, the combination of CLS-token-tuning, A-tuning, Affix-tuning, LoRA(out_proj), LoRA(in_proj), LoRA(dt_proj), $LoRA_p(X)$, and $LoRA_p(B)$ is the best.

| Step 1 | | Step 2 | |
|---|---|---|---|
| boolean search to use or not | fixed hyperparameters | fixed methods to use | hyperparameter search |
| CLS-token-tuning | wd=1e-3 | CLS-token-tuning | lr [1e-4, 5e-3] |
| Bias-tuning | wd=1e-3 | | wd [1e-5, 1e-2] |
| Pos-embed-tuning | wd=1e-3 | A-tuning | lr [1e-4, 5e-3] |
| D-tuning | wd=1e-3 | | wd [1e-5, 1e-2] |
| A-tuning | wd=1e-3 | Affix-tuning | n [1, 3] |
| Conv1d-tuning | wd=1e-3 | | lr [1e-4, 5e-3] |
| Prompt-tuning | n=1 | | wd [1e-6, 1e-3] |
| Affix-tuning | n=1 | LoRA(out_proj) | r [4, 16] |
| Additional-scan | n=1 | | s [1e-2, 1] |
| ParallelAdapter | r=8, s=0.1 | | lr [1e-4, 5e-3] |
| LoRA(embed) | r=8, s=0.1 | | wd [1e-6, 1e-3] |
| LoRA(x_proj) | r=8, s=0.1 | LoRA(in_proj) | r [4, 16] |
| LoRA(dt_proj) | r=4, s=0.1 | | s [1e-2, 1] |
| LoRA(out_proj) | r=8, s=0.1 | | lr [1e-4, 5e-3] |
| LoRA(in_proj) | r=8, s=0.1 | | wd [1e-6, 1e-3] |
| $LoRA_p(d)$ | r=4, s=0.1 | LoRA(dt_proj) | r [4, 16] |
| $LoRA_p(B)$ | r=4, s=0.1 | | s [1e-2, 1] |
| $LoRA_p(C)$ | r=4, s=0.1 | | lr [1e-4, 5e-3] |
| $LoRA_p(X)$ | r=8, s=0.1 | | wd [1e-6, 1e-3] |
| $LoRA_p(Z)$ | r=8, s=0.1 | $LoRA_p(X)$ | r [4, 16] |
| | | | s [1e-2, 1] |
| | | | lr [1e-4, 5e-3] |
| | | | wd [1e-6, 1e-3] |
| | | $LoRA_p(B)$ | r [4, 12] |
| | | | s [1e-2, 1] |
| | | | lr [1e-4, 5e-3] |
| | | | wd [1e-6, 1e-3] |
| | | select { | remove CLS-token-tuning |
| | | | remove A-tuning |
| | | | remove Affix tuning |
| | | | remove LoRA(out_proj) |
| | | | remove LoRA(in_proj) |
| | | | remove LoRA(dt_proj) |
| | | | remove $LoRA_p(X)$ |
| | | | remove $LoRA_p(B)$ |
| | | | not remove |

