# OpenReview forum: "MambaPEFT: Exploring Parameter-Efficient Fine-Tuning for Mamba"
_ICLR.cc/2025/Conference — ICLR 2025 Poster_

### Official Review · Reviewer_jdYa · 2024-10-23

**Soundness:** 2
**Presentation:** 3
**Contribution:** 2
**Rating:** 5
**Confidence:** 4

**Summary:**

This paper explores the PEFT methods for the Mamba. By modifying and adapting commonly used PEFT methods on Transformer, the paper proposes novel PEFT techniques tailored to the Mamba architecture. Experiments demonstrate that Mamba outperforms Transformer when applying PEFT methods. The paper also showcases how to effectively combine multiple PEFT methods to improve performance and introduces a new framework.

**Strengths:**

1 The authors extend PEFT methods from Transformer to the Mamba architecture and propose several Mamba-specific PEFT methods.

2 The authors conduct experiments not only in the vision domain but also in the language domain.

**Weaknesses:**

1 Why wasn’t Vim-B selected for comparison experiments with ViT-B? The authors seem to have not provided an explanation in the paper.

2 In Table 1, the PEFT methods selected by the authors are relatively outdated, such as FACT, which originates from AAAI23.

3 Conducting PEFT experiments on Vmamba, another Mamba-based vision model, is necessary to demonstrate the generality of the proposed methods.

4 In section 3.4, the mixed PEFT search strategy proposed by the authors lacks sufficient theoretical support, especially regarding the principles behind why different combinations of PEFT methods can improve performance, which the authors have not sufficiently explained.

5 For the vision experiments, the authors only consider smaller models without discussing the applicability of these methods to larger-scale models.

6 For the new architecture—Mamba—the training duration of these PEFT methods is worth considering and comparing.

**Questions:**

see Weaknesses.

---

> ### Author Response · Authors · 2024-11-26
> **Reply to Reviewer jdYa**
>
> Thank you for carefully reviewing our paper and providing valuable feedback. We will use it to improve the quality of our paper. Our responses to each of your concerns are as follows.
>
>
> ## [W1]
>
> We have added additional experiments with Vim-B, Vmamba-S and Vmamba-B. Please read the "Reply to all reviewers".
> The complete results of the experiments have been added to the appendix of the paper.
>
> ## [W2]
>
> To address the concern that Transformers were tested with relatively outdated baseline, we conducted additional experiments with the latest well-known method, SPT[1], and the state-of-the-art method, Adapter+[2], to ViT and added the experimental results to the Table 1 and Table 4. To the best of our knowledge, Adapter+ achieves the state-of-the-art accuracy on VTab-1K pre-trained with Imagene21K.
>
> [1]He, Haoyu, et al. "Sensitivity-aware visual parameter-efficient fine-tuning." Proceedings of the IEEE/CVF International Conference on Computer Vision. 2023.
> [2]Steitz, Jan-Martin O., and Stefan Roth. "Adapters Strike Back." Proceedings of the IEEE/CVF Conference on Computer Vision and Pattern Recognition. 2024.
>
> ## [W3]
>
> To further verify the generalizability of our method, we conducted additional experiments with VMamba. Below are extracts of the results.
>
> | base model | method |  #Params (K)  | Natural | Specialized | Structured | Average |
> |--------|------|------------|------------|------------|------------|-----------|
> | VMamba-S | Full | 49,379 | 66.4 | 81.8 | 62.6 | 68.0 |
> | | Linear | 18 | 67.9 | 73.6 | 34.2 | 54.9 |
> | | Conv2d-tuning | 160 | 75.8 | 83.3 | 58.9 | 70.3 |
> | | Affix-tuning(w/o proj) | 198 | 77.3 | 82.7 | 59.7 | 71.0 |
> | | Additional-scan | 395 | 76.4 | 83.3 | 60.2 | 71.0 |
> | | ParallelAdapter | 774 | 77.8 | 84.5 | 63.7 | 73.2 |
> | | LoRA(out proj) | 1,529 | 77.9 | **85.9** | **65.6** | **74.4** |
> | | LoRA$_p$(X) (= LoRA(in_proj)) | 1,529 | **78.2** | 84.5 | 65.2 | 74.0 |
> | VMamba-B | Full | 87,533 | 66.8 | 81.4 | 60.1 | 67.1 |
> | | Linear | 24 | 68.4 | 79.1 | 30.2 | 54.6 |
> | | Conv2d-tuning | 189 | 78.0 | 84.1 | 63.4 | 73.1 |
> | | Affix-tuning(w/o proj) | 276 | 77.5 | 83.3 | 59.2 | 71.0 |
> | | Additional-scan | 527 | 77.0 | 83.8 | 60.1 | 71.3 |
> | | ParallelAdapter | 1,032 | **78.6** | 85.1 | 63.3 | 73.5 |
> | | LoRA(out proj) | 2,039 | **78.6** | **85.7** | 64.9 | **74.3** |
> | | LoRA$_p$(X) (= LoRA(in_proj)) | 2,039 | 78.3 | 85.3 | **65.2** | **74.3** |
>
> Although VMamba improves the original Mamba architecture a lot as discussed in "Reply to all Reviewers," our method still works well with VMamba.
> The complete results of the experiments have been added to the appendix of the paper.
>
> ## [W4]
>
> While existing hybrid PEFT methods explore a combination of several promising PEFT methods and their hyperparameters, we conducted a greedy search of 20 PEFT methods and their hyperparameters. This led to new discoveries and improvements in accuracy, as discussed later. However, the search space was too vast to search simply. Our two-step searching approach significantly reduced the search space while enabling the search of the 20 PEFT methods. We will revise the paper to emphasize this point. Additionally, the main purpose of this experiment is to demonstrate the non-trivial result that combining PEFTs can improve performance. The experimental results further show the counterintuitive finding that combining weaker PEFTs yields higher performance than simply combining stronger PEFTs.
> The Table <Combining multiple LoRAs> in the reply for reviewer U29m should support this claim. Combining individually strong LoRAs did not improve accuracy.
>
> ## [W5]
>
> As mentioned in the reply for [W1] and [W3], we conducted additional experiments with Vim-B, VMamba, and their variations to address the concern about the lack of experiments with larger models.
>
> ## [W6]
>
> We added computation time cost information in Table 1. Note that the time cost of the embedding projection used for Affix-tuning, Prompt-tuning, and Hybrid PEFT is not large despite the parameter count. This is because, while other parameters process many tokens (for example, 16x16 in Vim), it only processes a few tokens. Moreover, if the batch size is 32, the computational cost is 1/32 compared to the other parameters since the input tokens to the embedding projection are data-independent parameters.
>
> Most of the time are consumed in the base Vim module, and the differences between the different PEFT methods are not significant.

---

> ### Comment · Reviewer_jdYa · 2024-11-27
>
> Thanks to the authors for their response and the additional experiments, which have addressed my concerns. I am willing to raise the score by one level.

---

> > ### Author Response · Authors · 2024-12-01
> >
> > Thank you very much for your positive feedback and increasing the score. We believe our paper is improved thanks to your constructive comments.

---

### Official Review · Reviewer_Czw3 · 2024-11-03

**Soundness:** 2
**Presentation:** 3
**Contribution:** 2
**Rating:** 6
**Confidence:** 4

**Summary:**

This paper explores parameter-efficient tuning methods such as the prominent methods like Adapter, LoRA, Prompt-tuning, and Partial tuning for the Mamba architecture. The authors highlight that these methods have not been studied in Mambas compared to the extensive research conducted on Transformers. The primary focus of this paper is on the Vision Mamba (Vim) architecture, with a minor portion addressing language models with Mamba compared with the Transformer counterparts Pythia. The study compares Vim and Vision Transformer models using PEFT methods, which address some key architectural and methodological choices like layer selection for LoRA and prompt placement in Prompt-tuning, for example. The authors present a two-step PEFT search method that hybridizes various PEFT options to offer a near-optimal solution (in terms of performance). Finally, the authors present a new PEFT method called Additional-scan, which increases the state dimension of the SSM, and an additional initialization method for the state matrix A is also presented. The experimental results employing Vim-S pre-trained on ImageNet-1K to demonstrate the effectiveness of the diverse PEFT options; Mamba-language models compared with Pythias are also presented.

**Strengths:**

- The paper is well-written and easy to follow.
- Searching for diverse PEFT options for Mamba is informative and would be a valuable contribution for those who employ Mamba architectures.
- Studying extensive PEFT options based on a baseline yields convincing results.

**Weaknesses:**

#### #  Main concerns
- The authors' claim in line 22 that PEFT is more effective for Mamba than for Transformers lacks adequate rationale and supporting evidence. There is no clear intuition as to why Mamba would benefit more from PEFT; the experiments appear biased due to an unfair comparison - specifically, Transformers were tested with only a limited range of PEFT options, while Mamba underwent more comprehensive testing (as illustrated in Table 1).
- The authors primarily tested Vision Mamba rather than the broader Mamba architecture, yet they frequently used "Mamba" as an abbreviation for Vision Mamba. This reviewer is concerned this may be misleading, especially as the result table for the language models offers only limited experiments. The authors should clarify their terminology to ensure readers understand that Vision Mamba is the specific variant.
- The authors presented a hybrid PEFT approach as an optimal-like solution, which is identified through their proposed search method. However, its conclusion does not specify a practical, generalizable option. This reviewer feels It's important to note that the effectiveness of the identified hybrid approach may not be universal, and the search process itself would be resource-intensive for a new architecture.

#### # Missing/wrong details in preliminary
- Section 2.1, titled *Mamba*, should provide foundational knowledge about Mamba as a preliminary, but it mainly presents on S4 or early SSMs. Since Mamba has slightly distinct notations and a signature architecture, the authors should revise this section to include more comprehensive details about Mamba.
- As the experiments involve the Vision Mamba architecture, the reviewer suggests including the original architecture, which has subtle design differences (e.g., the definition of B).
- The discretization presented in Eq.(2) is not the zero-order hold discretization as implied but rather the bilinear one. The authors should correct this detail.

####  # Some concerns about experiments
- The experiments used a limited set of baseline models. Although the PEFT options explored were diverse and met the experimental standards of a scientific study, only the Vim-S baseline is used (since the PEFT methods are more likely to be applied to larger architectures, Vim-S may not be a proper choice as a baseline), where this reviewer feels the main comparison focuses on Vision Mambas versus Vision Transformers.
- Vision Transformers are not thoroughly explored compared to the fully examined Mamba models. For instance, ViT-S in Table 1 has only a few options compared to Mamba's more comprehensive use of PEFTs.
- Some experimental comparisons lacked control:
  - In language models, for instance, Pythia 1.4B and the counterpart Mamba 1.4B had significantly different baseline performances (44.8 vs. 53.0), skewing the reported improvements, as higher starting accuracy inflates gains.

#### # Minors
- There is a related work that covers a similar topic, including PEFTs in Mamba, which suggests that this paper is not the first to study PEFT for the Mamba architecture as mentioned in lines 71-74. This reviewer understands that the authors could not include the related work due to its publication date being after this submission. However, it is suggested to compare this study with that work in future revisions to provide context and emphasize differences.
  - Parameter-Efficient Fine-Tuning of State Space Models, NeurIPS 2024 workshop (https://arxiv.org/abs/2410.09016)

- Typo in Table 1: Linear 'Proving'

**Questions:**

- Could the authors provide additional results using Vim-Base or larger-scale architectures?
- How was the data utilized in the hybrid search method? Was a held-out set created for the searches?

---

> ### Author Response · Authors · 2024-11-26
> **Reply to Reviewer Czw3 (About Main concers)**
>
> Thank you for carefully reviewing our paper and providing insightful feedback. We will use it to improve the quality of our paper. Our responses to each of your concerns and questions are as follows.
>
>
> # [Main concerns]
>
>
> ## [1]
>
> To address the concern that Transformers were tested with a limited set of PEFT methods, we conducted additional experiments. We applied the latest well-known method, SPT[1], and the state-of-the-art method, Adapter+[2], to ViT and added the experimental results to Table 1 and Table 4. To the best of our knowledge, Adapter+ achieves state-of-the-art accuracy on VTab-1K pre-trained with Imagene21K. Therefore, while the number of PEFT methods for ViT in Table 1 is limited, we believe we have made a fair comparison by selecting the state-of-the-art methods from a vast number of existing methods.
>
> The fact that PEFTs for Mamba show a larger improvement from conventional full fine-tuning and linear probing than the PEFTs for ViT do supports our claim that Mamba benefits more from PEFT than Transformers. Additionally, to facilitate future research, we have chosen and proposed simple designs for all Mamba PEFTs. Despite this, our methods outperform even the most accurate ViT PEFTs. Proposing more complex PEFT designs to further improve accuracy is future work.
>
> [1]He, Haoyu, et al. "Sensitivity-aware visual parameter-efficient fine-tuning." Proceedings of the IEEE/CVF International Conference on Computer Vision. 2023.
> [2]Steitz, Jan-Martin O., and Stefan Roth. "Adapters Strike Back." Proceedings of the IEEE/CVF Conference on Computer Vision and Pattern Recognition. 2024.
>
>
> ## [2]
>
> As the reviewer pointed out, there were several places where the distinction between the original Mamba, Mamba for visual tasks, and its variant Vim was confused. In order to make the distinction clearer and to better convey our intentions, we have updated the paper.
>
>
> ## [3]
>
> Our HybridPEFT has been evaluated on all tasks included in VTAB. In this respect, it generalizes better compared to NAS-based method[3] that specialize on specific tasks. Unlike some existing works, we did not perform tuning for specific tasks in VTAB-1k. Similar to many Hybrid PEFT methods, our approach also requires retraining and exploration when there are changes in the model architecture or PEFT methods.
>
> Through Hybrid PEFT exploration, we validated the non-trivial hypothesis that combining PEFTs with different mechanisms can improve performance. In addition, we obtained the counterintuitive hypothesis that combining weaker PEFT methods yields higher performance than simply combining stronger PEFT methods.
>
> Regarding the computational cost of the Hybrid PEFT, as stated in the 'HYBRID PEFT SEARCH' section of the appendix, each trial takes only around 20 minutes on a single GPU. Our approach, which dramatically reduces the search space by dividing it into two steps, converges well with a total of 200 trials, requiring only about 70 GPU hours. It is easy to optimize for new model structures with a few GPUs.
>
> [3]Zhang, Yuanhan, Kaiyang Zhou, and Ziwei Liu. "Neural prompt search." arXiv preprint arXiv:2206.04673 (2022).

---

> > ### Author Response · Authors · 2024-11-26
> > **Reply to Reviewer Czw3 (About other points)**
> >
> > # [Missing/wrong details in preliminary]
> >
> > 1. We will accurately describe Mamba in the Prelimitation section and update the text to make it self-contained.
> >
> > 2. Our understanding is that the B in Mamba and Vision Mamba is the same. We corrected the Prelimitation section as it was misleading.
> >
> > 3. We corrected Equation (2). Thank you for pointing it out.
> >
> >
> > # [Some concerns about experiments]
> >
> > ## [1]
> >
> > In addition to Vim-S and Vim-tiny, we have added experiments with Vim-B, Vmamba-S and Vmamba-B. Please read the "Reply to all reviewers".
> >
> >
> > ## [2]
> >
> > As mentioned in reply for main concern-1, we conducted additional experiments on Vision Transformer using the latest well-known method, SPT, and the state-of-the-art method, Adapter+. These additional experiments will allow for a fairer comparison between the results of Vision Transformer and Mamba.
> >
> > ## [3]
> >
> > As stated in the experimental setup of the appendix, we could adjust the learning rate for PEFT method per model size thanks to its lower computational cost. However, since full fine-tuning has a higher computational cost, we reused the adjusted learning rate for smaller models. We have updated Table 3 with the results after adjusting the learning rates for full fine-tuning. The results of these additional experiments do not affect our claims.
> >
> > # [Minors]
> > ## [1]
> >
> > We will update final version of our paper to include references which published after the submission.
> >
> > Regarding the paper exemplified by the reviewer: Galim et al. pointed out the expressiveness limitations of Prefix-tuning in SSM, as it only tunes the initial state. Our proposed Affix-tuning can deal with this issue due to its algorithmic nature, which allows tokens to be inserted at arbitrary positions. Actually, Affix-tuning has demonstrated superior performance to LoRA in language tasks.
> >
> >
> > ## [2]
> >
> > Thank you for pointing out the typo. We have corrected the typos and formatting in the paper, including the point you mentioned.
> >
> >
> > # [Q1]
> >
> > As mentioned in our response to [Some concern about experiment-1], we conducted additional experiments with Vim-B, VMamba, and their variations to address the concern about the lack of experiments with larger models.
> >
> >
> > # [Q2]
> >
> > The Hybrid PEFT search is optimized based on the accuracy of the training set using 5 datasets from VTAB-1k. The evaluation is conducted using the test sets of 19 datasets. Thus we think that the risk of overfitting is minimal in our problem setting of selecting PEFT methods. Furthermore, the hyperparameters for all PEFT methods are optimized to maximize accuracy, similar to the Hybrid PEFT, which allows for a fair comparison. Please note that we use the same hyperparameters across all tasks to provide generally useful information, while some prior studies tune hyperparameters for each task.

---

> ### Comment · Reviewer_Czw3 · 2024-11-30
> **Official comment by the Reviewer Czw3**
>
> Thank you for your time and effort in providing detailed responses and conducting additional experiments. Though some of my concerns are still unaddressed, I acknowledge the authors' significant effort during the rebuttal period and have decided to increase the rating.
>
>
> Aside from the increased rating, this reviewer woud like to encourage the authors to enhance the paper by addressing the remaining issues. Agreeing with the other reviewers' concerns regarding novelty and scope, this reviewer recommends the authors revise the paper more to focus on its key contributions with giving a clearer intution. The response on intuition seems to be based primarily on experimental results. However, from my perspective, it would be more effective to provide some intuitive  conjectures, even if they are informal; then, the experimental results could support them afterwards.
>
> Furthermore, the experimental designs could be more refined, which should ensure a closer apples-to-apples comparison. For example, Table 3 should include more experiments for Pythias, similar to the Mamba cases, and the baseline accuracies should be aligned like the aligned number of parameters. Specifically, though the further provided experiments address more cases as requested, similar experiments could be done with Transformers by applying PEFTs on individual layers, like embed or out_proj, rather than grouping them all together, as in Mamba's cases. Therefore, the experimental studies on Mamba generally involved more effort compared to Transformers and employed unaligned baselines, which weaken the main claim: "Our experiments indicate that PEFT performs more effectively for Mamba than Transformers".

---

> ### Comment · Reviewer_Czw3 · 2024-12-02
> **Please check my comment which was not visible to the authors.**
>
> Thank you for your time and effort in providing detailed responses with additional experiments. I have updated the final rating, as you may have noticed. Please also check my comment, which I realized was invisible to the authors earlier; now it has been fixed.

---

> > ### Author Response · Authors · 2024-12-03
> >
> > Thank you very much for your positive feedback and increasing the score. We also appreciate your additional advice, which we can now read. In the final version, we will take your valuable advice into consideration: focusing clearly on our key contributions and providing more evaluation on Physia.

---

### Official Review · Reviewer_U29m · 2024-11-03

**Soundness:** 3
**Presentation:** 2
**Contribution:** 3
**Rating:** 8
**Confidence:** 5

**Summary:**

This paper explores the application of parameter-efficient fine-tuning (PEFT) in the Mamba model. It begins by systematically analyzing the suitability of existing PEFT methods for the Mamba model, then adapts certain PEFT methods to better align with the characteristics of State Space Models (SSMs), and finally investigates hybrid PEFT search strategies. Extensive experiments were conducted across image and text modalities, providing thorough benchmarks for various PEFT methods and demonstrating that the Mamba model exhibits distinct fine-tuning behavior compared to Transformer models.

**Strengths:**

1. The paper focuses on a novel and competitive model architecture, SSMs, systematically analyzing the applicability of existing PEFT methods for the Mamba model, offering insights for designing efficient fine-tuning schemes for SSMs.
2. The paper adapts existing PEFT methods specifically for SSMs, proposing new approaches tailored for Mamba, achieving good performances.
3. The extensive experiments, covering both image and text modalities, provide comprehensive benchmarks for all methods and reveal differences in PEFT performance between the Mamba and Transformer models.

**Weaknesses:**

1. The paper covers too many aspects, including various PEFT methods, individual analyses, and hybrid architecture search. The authors should concentrate on one aspect and provide valuable conclusions for each part instead of describing each in detail, as this could reduce readability and divert focus.
2. The paper should consider including the most advanced works within each PEFT category, such as GPS[1] and SPT[2] for partial-tuning methods, and SSF[3] in addition to LoRA-based methods for reparameterization approaches.
3. Experiments should involve larger-scale models, where PEFT methods are particularly valuable. Evaluation across different model scales is crucial for PEFT methods. Notably, "scale" here refers to model parameter size rather than training data size (e.g., a model trained on ImageNet-21K in Section 4.5 is not necessarily larger than one trained on ImageNet-1K; model parameter size is more relevant in this context).
3. Presentation quality requires improvement, including spelling errors (e.g., "Linear Probing" instead of "Linear Proving" in Table 1’s ViT-S section) and inconsistent heights in sub-tables within Table 2.

[1] Zhang Z, Zhang Q, Gao Z, et al. Gradient-based Parameter Selection for Efficient Fine-Tuning[C]. Proceedings of the IEEE/CVF Conference on Computer Vision and Pattern Recognition. 2024: 28566-28577.

[2] He H, Cai J, Zhang J, et al. Sensitivity-aware visual parameter-efficient fine-tuning[C]. Proceedings of the IEEE/CVF International Conference on Computer Vision. 2023: 11825-11835.

[3] Lian D, Zhou D, Feng J, et al. Scaling & shifting your features: A new baseline for efficient model tuning[J]. Advances in Neural Information Processing Systems, 2022, 35: 109-123.

**Questions:**

1. Table 1 shows that Adapter-based and LoRA-based methods perform best. Did the authors try SequentialAdapter, or, as in the original LoRA paper, apply LoRA across multiple locations simultaneously in the model?
2. In Table 1, the Hybrid approach shows no significant advantage over individual PEFT methods (e.g., ParallelAdapter or LoRA) despite having far more parameters (117,236). The performance of Hybrid is not markedly better than LoRA(in_proj), which has only 1,483 learnable parameters.
3. Beyond parameter efficiency, computational efficiency is also crucial for PEFT methods. Did the authors consider comparing the computational efficiency of different methods (e.g., in terms of FLOPs or running time)?

---

> ### Author Response · Authors · 2024-11-26
> **Reply to Reviewer U29m**
>
> Thank you for carefully reviewing our paper and providing insightful feedback. We will use it to improve the quality of our paper. Our responses to each of your concerns and questions are as follows.
>
>
> ## [W1]
>
> We acknowledge that we have conducted extensive explorations with many proposal, leading to a wide range of new discoveries and novelties that warrant discussion. We summarized them clearly in the Conclusion Section. Do you think this makes the new discoveries and novelties easier to understand?
>
>
> ## [W2]
>
> Thank you for your suggestion. We included SPT, GPS, and SSF in the related work section. We also did experiments of SPT with ViT and added in Table 1 and Table 3. We will also add SSF and GPS if we finish the experiments. On the other hand, We will refrain from experimenting with Vim this time. The reason is that, as pointed out in [W1], this paper already covers many aspects, and including more than two methods in each category (e.g., reparameterization method) would complicate the discussion further. Additionally, SPT and GPS involve a two-step training process and require different handling compared to other methods. It would make the discussion of performance factors in Hybrid PEFT more challenging.
> By using simple methods (e.g., LoRA, ParallelAdapter, Affix-tuning), we aim to minimize extraneous influences, allowing evaluation results to be universally informative for various future tuning strategies.
>
>
> ## [W3]
>
> We have added additional experiments with Vim-B and Vmamba-B, models with a larger number of parameters. Please read the "Reply to all reviewers".
>
>
> ## [W4]
> Thank you for pointing this out. We have addressed the issue including other parts we noticed.
>
>
> ## [Q1]
>
> We tried SequentialAdapter, the original Adapter, to Vim but ParallelAdapter showed better accuracy similar to the case for ViT. That's why we used the ParallelAdapter in our experiments. We also applied LoRA across multiple locations simultaneously. As shown in the table below, combining individually high-accurate LoRAs, LoRA(in_proj)+LoRA(out_proj) or combining all LoRAs resulted in lower accuracy than partial LoRA(X). This suggests that applying LoRA only to important features, like Partial LoRA(X), is effective. Furthermore, to pursue even higher accuracy, it is crucial to find good combinations of PEFT methods, as proposed in the hybrid PEFT, rather than simply combining individually high-accurate LoRAs. We will add these results to the paper to strengthen our claims.
>
> **<Combining multiple LoRAs>**
> | method | rank | VTab Avg. |
> |--------|------|------------|
> | LoRA(in_proj)+LoRA(out_proj)  | 8   | 71.33 |
> | LoRA(in_proj)+LoRA(out_proj)  | 16   | 70.68 |
> | LoRA(in_proj)+LoRA(out_proj)  | 32   | 69.77 |
> | all LoRAs  | 8 | 71.02 |
> | all LoRAs  | 16 | 70.42 |
> | all LoRAs  | 32 | 68.74 |
> | LoRA$_p$(X)  | 64 | **71.52** |
>
>
> ## [Q2]
> The reason the number of parameters in Hybrid PEFT is large is simply because the embedding projection in Affix-tuning has a large number of parameters. However, as explained later in [Q3], the computational cost is almost the same with others. If there is sufficient GPU memory, it can be trained quickly as well, and during inference, the embedding projection can be merged within the embedding weights. Therefore, it will be a good choice to choose Hybrid PEFT when pursuing accuracy. Moreover, the Hybrid PEFT (wo_proj), which does not use the embedding projection, achieves an accuracy of 71.8 with 1,044K parameters, outperforming LoRA(in_proj) in both parameter count and accuracy. Furthermore, our finding that it is important to find good combinations rather than just combining individually good PEFT methods, supported further by additional experiments in [Q1], is another contribution of our Hybrid PEFT.
>
>
> ## [Q3]
> We added computation time information in Table 1. Note that the time cost of the embedding projection used for Affix-tuning, Prompt-tuning, and Hybrid PEFT is not large despite the parameter count. This is because, while other parameters process many tokens (for example, 16x16 in Vim), it only processes a few tokens. Moreover, if the batch size is 32, the computational cost is 1/32 compared to the other parameters since the input tokens to the embedding projection are data-independent parameters.
>
> Most of the time are consumed in the base Vim module, and the differences between the different PEFT methods are not significant.

---

> > ### Comment · Reviewer_U29m · 2024-11-27
> >
> > I would like to thank the authors for their efforts and detailed responses, which have largely addressed my concerns. The work on exploring PEFT for the Mamba architecture is valuable, ao I have raised my score to 8.

---

> > > ### Author Response · Authors · 2024-12-01
> > >
> > > Thank you very much for your precious feedback and increasing your score. We believe our paper is improved thanks to your constructive comments.

---

### Official Review · Reviewer_hvVY · 2024-11-07

**Soundness:** 2
**Presentation:** 2
**Contribution:** 2
**Rating:** 5
**Confidence:** 4

**Summary:**

This paper presents a comprehensive and detailed exploration of parameter-efficient fine-tuning (PEFT) for Mamba. By applying and refining existing PEFT methods, introducing novel PEFT approaches（Partial-tuning and Additional-scan）, and exploring optimal algorithmic combinations, it achieves outstanding performance in both image and language tasks. This work provides valuable insights for efficient fine-tuning in Mamba.

**Strengths:**

1. The paper not only improves existing PEFT algorithms to suit Mamba but also proposes targeted, innovative algorithms, providing the community with a broader selection of PEFT methods.
2. The experiments comprehensively showcase the performance of each PEFT algorithm, offering a framework for optimal algorithmic combinations.
3. The extensive experimental workload demonstrates thorough exploration of the algorithms.

**Weaknesses:**

1. The improved and original PEFT algorithms did not achieve the best results. In the VTAB-1k benchmark, the performance of the improved Partial LoRA is nearly identical to that of LoRA, while Additional-scan underperforms ParallelAdapter with comparable training parameters. Compared to traditional PEFT algorithms, these methods lack strong competitive advantage.
2. Although LoRA demonstrates outstanding performance on the VTAB-1k benchmark, its comparison with the modified Partial LoRA on language tasks is missing, making it unclear whether Partial LoRA is indeed the optimal PEFT method.
3. The algorithmic novelty is limited, as the concept of Partial-tuning appears relatively straightforward.

**Questions:**

Please further explain the advantage and effectiveness of the proposed method compared with the baselines.
Please further explain the novelty of the proposed method.

---

> ### Author Response · Authors · 2024-11-26
> **Reply to Reviewer hvVY**
>
> Thank you very much for your constructive review and valuable suggestions. Below, you will find detailed responses to your questions and comments. Please let us know if our answers do not fully address your concerns. We will respond to you immediately.
>
>
> ## [W1 and W3]
>
> We acknowledge that Partial LoRA has limited novelty. Our goal is to explore PEFT for Mamba—specifically, how to tune and where to tune. By using a simple LoRA, we aim to minimize extraneous influences, allowing evaluation results to be universally informative for various future tuning strategies. During the discussion period, we noticed that a concurrent work of Lily [1], which improved from LoRA, added experiments with Vim on 26 Sep. 2024. Our Partial LoRA(X) achieved 1.8 points higher accuracy than Lily, highlighting the importance of detailed exploration of where to tune even partially and how to tune with suitable number of parameters. (Please compare individual task accuracies, as averaging ways differ.)
>
> Conversely, we believe Affix-tuning and Additional-scan possess significant novelty and usefulness, despite their intended simple design. They provide a lighter weight and higher accuracy than LoRA in language tasks. As hypothesized in Section 4.4, we attribute this to the volume of training data. To support this, we conducted experiments increasing training data from 1,000 to 20,000 on VTabl-1K, using official test data as training data and vice versa. The results of the Retinopathy task are below.
>
> **<The results with increased training data>**
> |                        | #param [K] | 1000 | 2000 | 5000 | 10000 | 20000 |
> |-----------------------|------------|------|------|------|-------|-------|
> | Affix-tuning (wo_proj) | **230** | 73.7 | 71.4 | 74.2 | **76.6**  | **77.3**  |
> | Additional-scan        | 673 | 70.3 | 73.0 | 72.4 | 73.3  | 77.0  |
> | LoRAp(X)               | 1778 | **74.2** | **73.7** | **75.2** | 76.5  | 76.7  |
>
> Further results across other tasks are included in Fig. 7. Based on these findings, we conclude that Affix-tuning and Additional-scan are efficient methods with a relatively large training dataset.
>
> Pursuing accuracy through complex design may risk the versatility and usefulness of an exploration work. Therefore, we would like to consider further improvements in accuracy through complex design as future work.
>
> [1] Zhong, Yibo, and Yao Zhou. "Low-Rank Interconnected Adaptation across Layers." arXiv preprint arXiv:2407.09946 (2024).
>
>
>
> ## [W2]
>
> Thank you very much for point it out. It will strengthen our paper. The results are below.
>
> | base model | method | avg. |
> |--------|------|------------|
> | Mamba 130M | LoRA(in_proj) | 42.8 |
> | | LoRA$_p$(X) | **43.7** |
> | Mamba 370M | LoRA(in_proj) | 47.2 |
> | | LoRA$_p$(X) | **47.8** |
> | Mamba 790M | LoRA(in_proj) | 50.7 |
> | | LoRA$_p$(X) | **50.8** |
> | Mamba 1.4B | LoRA(in_proj) | 52.6 |
> | | LoRA$_p$(X) | **53.7** |
>
> Partial LoRA (X) achieved higher accuracy in language tasks as well. The detailed results were added in Table 3.
>
>
>
> ## [Q1]
>
> As discussed in [W1 and W3], we believe Affix-tuning and Additional-scan possess significant novelty and usefulness. However, we realized after receiving your feedback that our writing did not convey the message adequately. Therefore, we revised the paper to clarify novelty and usefulness.
>
> As to the Partial LoRA, the novelty is limited, since we used the simple LoRA. However, we believe we were able to systematically discover which parts should be tuned with how many trainable parameters thanks to the simplicity. Based on our findings, future work can focus on effective PEFT research without searching a lot.
> For example, if we tinker partial LoRA(X) with the latest DoRA instead of LoRA, the accuracy increased as below.
>
> |  | VTAB-1K with Vim-S  | Commonsense reasoning with Mamba 1.4B |
> |--------|------|------------|
> | LoRA$_p$(X) | 71.5 | 53.7 |
> | DoRA$_p$(X) | 71.6 | 53.9 |
>
> The more detail can be found in Table 6.

---

### Author Response · Authors · 2024-11-26
**Reply to all Reviewers**

Thank you for carefully reviewing our paper and providing valuable feedback. Bellow are for all Reviewers.

# Some updates of the results
We made some corrections regarding the results of the paper, as there were some mistakes and issues with the training settings. One correction is that the accuracy of Affix-tuning in Table 1 was incorrectly reported as too low; the correct values were in Appendix Table 4. Additionally, the significantly lower accuracy of Affix-tuning (wo_proj) and Prompt-tuning (wo_proj) compared to Affix-tuning and Prompt-tuning is due to the learning rate being too small for the embedding module. Other PEFT parameters were either over-parameterized with DNN or fine-tuned from already trained values, which allowed us to train them sufficiently within 100 epochs. However, the parameters in the embedding module are not over-parameterized and require tuning from scratch. Therefore, to converge the training within 100 epochs, the learning rate needed to be increased by 10 times. We have updated Table 1 and Table 4 with the improved values.

# Experiments with Vim-B
We received a concern about the lack of evaluation on larger models from three reviewers. So we will discuss it here.
To address the concern regarding the lack of experiments with Vim-B, we conducted additional experiments. The results of Vim-B were not included initially because the model information was not publicly available at the time of our experiments (this August).
Below are extracts of the results for comparison between Vim-S and Vim-B.

| base model | method |  #Params (K)  | Natural | Specialized | Structured | Average |
|--------|------|------------|------------|------------|------------|-----------|
| Vim-S | Full | 25,450 | 59.4 | 68.7 | 34.4 | 47.1 |
| | Linear | 9 | 62.5 | 77.3 | 32.0 | 52.8 |
| | Affix-tuning (w/o proj) | 230 | 63.9 | 77.7 | 34.2 | 54.3 |
| | Affix-tuning | 117,000 | 75.8 | 83.3 | 58.9 | 70.3 |
| | Additional-scan | 672 | 74.6 | 82.7 | 56.4 | 68.7 |
| | ParallelAdapter | 663 | 76.1 | 84.0 | 60.0 | 71.0  |
| | LoRA(out_proj) | 2,663 | 76.4 | 84.0 | 60.1 | 71.1 |
| | LoRA(in_proj) | 1,483 | **76.6** | **84.1** | 60.2 | 71.3 |
| | LoRA$_p$(X) | 1,778 | **76.6** | 83.9 | **60.8** | **71.5** |
| Vim-B | Full | 96,906 | 43.9 | 69.5 | 27.8 | 42.5 |
| | Linear | 18 | 64.5 | 79.2 | 32.2 | 54.0 |
| | Affix-tuning (w/o proj) | 460 | 69.2 | 82.2 | 50.8 | 64.2 |
| | Affix-tuning | 463,000 | 76.4 | 83.5 | 55.6 | 69.1 |
| | Additional-scan | 690 | 75.5 | 82.4 | 55.8 | 68.7 |
| | ParallelAdapter | 1,787 | 76.6 | 83.9 | **60.4** | **71.3** |
| | LoRA(out_proj) | 5,916 | 76.7 | 84.0 | 60.1 | **71.3** |
| | LoRA(in_proj) | 2,987 | 76.7 | **84.3** | 59.4 | 71.0 |
| | LoRA$_p$(X) | 3,557 | **77.0** | 83.9 | 59.9 | **71.3** |

From Vim-S to Vim-B, there is little improvement, whereas ViT-B improved from ViT-S. Because this phenomenon is not observed in vanilla Mamba for language tasks (see Table 3), it seems it is not due to PEFT methods for Mamba. This suggests that large Vim/Mamba models might face potential challenges when fine-tuning with limited data. Note that even with Vim-B, PEFT still significantly outperforms full fine-tuning and achieves equivalent or higher accuracy than state-of-the-art PEFT with ViT-B, indicating that PEFT for Vim remains effective.

We also conducted experiments with VMamba-S and VMamba-B [1] to check the generalizability. The results are shown in the reply to the reviewer jdYa and added to the appendix. VMamba made various improvements over the original Mamba. It eliminated the state dimension (in other words, reduced the state dimension to 1), introduced a hierarchical structure similar to CNNs, replaced causal 1D convolutions with 2D convolutions, removed the Gated MLP structure, and substituted half of the Mamba blocks with MLP blocks. Due to these changes, there are some differences in trends, but the proposed method works also well with VMamba, achieving higher performance on the VTAB-1K dataset than Vim as the original VMamba performs better on ImageNet1K evaluation.

[1] Yue Liu, Yunjie Tian, Yuzhong Zhao, Hongtian Yu, Lingxi Xie, Yaowei Wang, Qixiang Ye, and Yunfan Liu. Vmamba: Visual state space model. arXiv preprint arXiv:2401.10166, 2024b.

---

### Meta-Review · Area_Chair_WvD8 · 2024-12-22

**Metareview:**

a) This paper explores the application of parameter-efficient fine-tuning (PEFT) for Mamba. It analyzes the suitability of existing PEFT methods for Mamba, adapts certain PEFT methods to better align with the characteristics of State Space Models (SSMs), and investigates hybrid PEFT search strategies. Extensive experiments on image and text modalities for various PEFT methods show that Mamba exhibits distinct fine-tuning behaviour compared to Transformers.

b) The paper improves existing PEFT algorithms to suit Mamba but also proposes targeted algorithms, providing the community with a broader selection of PEFT methods. The extensive experiments, covering both image and text modalities, provide comprehensive benchmarks for all methods and reveal differences in PEFT performance between the Mamba and Transformer models.

c) The algorithmic novelty of the paper is limited, as the proposed approaches are not very different form standard PEFT. For some results (e.g. PEFT is more effective for Mamba than for Transformers), authors provide empirical evidence but not theory, a rationale or intuition about the results.

d) Overall, I consider that the paper deserves publication because:
PEFT methods for Mamba architectures are still under-explored. The authors provided additional experimental results in their rebuttal, which addressed most of the concerns raised by reviewers.
The paper includes extensive experiments covering various PEFT methods, different Mamba architectures, models of different scales, and different tasks, which can be have a strong impact for further research in the field.
However, I still think authors should work on further improving the presentation of the paper, for instance clarifying the contribution and providing a better understanding of the empirical results obtained, as suggested by reviewers.

**Additional Comments On Reviewer Discussion:**

Rev. hvVY scored the paper with 5. He appreciate the presented methodology and experiments but did not find enough clarity on the novelty of the approach and the advantage compared to baselines. Authors answered to all questions, but rev. did not engage in further discussions.

Rev. U29m appreciate how the PEFT are adapted for Mamba and the extensive experiments, but found also many weaknesses. Authors rebuttal was appreciated and the score was raised to 8.

Rev. Czw3 found the paper well written and useful for Mamba users. However, he provided an extensive list of comments and weaknesses. Authors answered to most of them and rev. was satisfied, suggested some additional improvements and increased their score to 6.

Rev. jdYa was the most negative, pointing out many limitation or missing points on the experiments. After rebuttal rev. liked the authors answers and increased their score to 5.

While scores are ranging from 5 to 8, the general understanding of the paper was very similar for all reviewer. The tackled problem is quite new, and although the techniques used are not very novel, results are good and helpful for the community. I appreciate the thorough review of Czw3 and I share their opinion on the overall considerations of the paper. I could not find any major problem that would imply rejection, however authors should work on further improving the presentation of the paper as suggested by rev. Czw3.

---

### Decision · Program_Chairs · 2025-01-22

Accept (Poster)